# Human cytomegalovirus antagonizes activation of Fcγ receptors by distinct and synergizing modes of IgG manipulation

Philipp Kolb[1,2], Katja Hoffmann[1,2], Annika Sievert[1,2], Henrike Reinhard[3], Eva Merce-Maldonado[3], Vu Thuy Khanh Le-Trilling[4], Anne Halenius[1,2], Dominique Gütle[1,2], Hartmut Hengel[1,2]*

[1]Faculty of Medicine, Albert-Ludwigs-University Freiburg, Freiburg, Germany; [2]Institute of Virology, University Medical Center, Albert-Ludwigs-University Freiburg, Freiburg, Germany; [3]Institute of Virology, University Hospital Düsseldorf, Heinrich-Heine-University Düsseldorf, Düsseldorf, Germany; [4]Institute for Virology, University Hospital Essen, University of Duisburg-Essen, Essen, Germany

**Abstract** Human cytomegalovirus (HCMV) is endowed with multiple highly sophisticated immune evasion strategies. This includes the evasion from antibody mediated immune control by counteracting host Fc-gamma receptor (FcγR) mediated immune control mechanisms such as antibody-dependent cellular cytotoxicity (ADCC). We have previously shown that HCMV avoids FcγR activation by concomitant expression of the viral Fc-gamma-binding glycoproteins (vFcγRs) gp34 and gp68. We now show that gp34 and gp68 bind IgG simultaneously at topologically different Fcγ sites and achieve efficient antagonization of host FcγR activation by distinct but synergizing mechanisms. While gp34 enhances immune complex internalization, gp68 acts as inhibitor of host FcγR binding to immune complexes. In doing so, gp68 induces Fcγ accessibility to gp34 and simultaneously limits host FcγR recognition. The synergy of gp34 and gp68 is compelled by the interfering influence of excessive non-immune IgG ligands and highlights conformational changes within the IgG globular chains critical for antibody effector function.

*For correspondence:
hartmut.hengel@uniklinik-freiburg.de

Competing interests: The authors declare that no competing interests exist.

## Introduction

Human cytomegalovirus (HCMV) constitutes the prototypical human pathogenic β-herpesvirus found worldwide with high immunoglobulin G (IgG) sero-prevalence rates of 56–94% depending on the respective countries (*Zuhair et al., 2019*). A hallmark of cytomegalovirus infection is the establishment of a lifelong persistence with recurring phases of latency and reactivation of productive infection and horizontal spread in presence of adaptive immune responses. HCMV encodes the largest known transcriptome of human viruses (*Stern-Ginossar et al., 2012*), giving rise to an equally large antigenic proteome including a huge and varied arsenal of immunoevasins (*Berry et al., 2020*; *Hengel et al., 1998*) that counteract immune recognition of infected cells facilitating virus persistence, shedding, and superinfection of sero-positive hosts. While primary HCMV infection of healthy individuals usually remains undetected, it can cause severe symptoms in the immunocompromised. Besides antiviral therapy involving nucleotide analogs, concentrated HCMV-immune IgG preparations (e.g. Cytotect) are used to prevent infection of immunocompromised patients including the prevention of congenital infection in primary infected pregnant women with varying degrees of success (*Kagan et al., 2019*; *Revello et al., 2014*). Generally, HCMV is observed to withstand a humoral immunity even replicating and disseminating in the presence of highly neutralizing immune sera in vitro and clinical isolates of HCMV tend to disseminate via cell-to-cell spread, thus avoiding an encounter with neutralizing antibodies (*Falk et al., 2018*). This, however, cannot fully explain the

**eLife digest** Human cytomegalovirus is a type of herpes virus that rarely causes symptoms in healthy people but can cause serious complications in unborn babies and in people with compromised immune systems, such as transplant recipients.

The virus has found ways to successfully evade the immune system, and once infected, the body retains the virus for life. It deploys an arsenal of proteins that bind to antibodies, specialized proteins the immune system uses to flag virus-infected cells for destruction. This prevents certain cells of the immune system, the natural killer cells, from recognizing and destroying virus-infected cells.

These immune-evading proteins are called viral Fc-gamma receptors, or vFcγRs. While it has been previously shown that these receptors are able to evade the immune system, it remained unknown how exactly they prevent natural killer cells from recognizing infected cells.

Now, Kolb et al. show that the cytomegalovirus deploys two vFcγRs called gp34 and gp68, which work together to block natural killer cells. The latter reduces the ability of natural killer cells to bind to antibodies on cytomegalovirus-infected cells. This paves the way for gp34 to pull virus proteins from the surface of the infected cell, making them inaccessible to the immune system. Neither protein fully protects virus-infected cells on its own, but together they are highly effective.

The experiments reveal further details about how cytomegalovirus uses two defense mechanisms simultaneously to outmaneuver the immune system. Understanding this two-part viral evasion system may help scientists to develop vaccines or new treatments that can protect vulnerable people from diseases caused by the cytomegalovirus.

resistance of HCMV to a humoral response leading to virus dissemination across several organs. Notably, HCMV encodes a set of Fcγ-binding glycoproteins (viral FcγRs, vFcγRs) that have been shown to antagonize host FcγR activation by immune IgG (*Corrales-Aguilar et al., 2014b*). In this previous study, we showed that HCMV gp34, gp68 and HSV-1 gE/gI efficiently antagonize the activation of human FcγRs. By attacking the conserved Fc part of IgG, HSV-1 and HCMV are able to counteract potent antiviral immune responses including antibody-dependent cellular cytotoxicity (ADCC). As it has become increasingly evident that Fcγ mediated immune control is essential not only for the effectiveness of non-neutralizing but also neutralizing IgG antibodies (*DiLillo et al., 2014*; *Forthal et al., 2013*; *Horwitz et al., 2017*; *Van den Hoecke et al., 2017*), a mechanistic and causal analysis of this evasion process is highly warranted in the pursuit of better antibody based intervention strategies targeting herpesviruses and HCMV in particular. While several herpesviruses encode vFcγRs, HCMV is the only virus known so far that encodes more than one individual molecule with the capacity to bind Fcγ. Specifically, HCMV encodes four distinct molecules which share this ability: gp68 (*UL119-118*), gp34 (*RL11*), gp95 (*RL12*), and gpRL13 (*RL13*) (*Atalay et al., 2002*; *Corrales-Aguilar et al., 2014a*; *Cortese et al., 2012*; *Sprague et al., 2008*). Turning to other species, mouse CMV (MCMV) encoded m138 has been shown to bind Fcγ, yet it has more prominently been associated with a variety of unrelated functions proving m138 not to be a strict homolog of the HCMV counterparts (*Arapović et al., 2009*; *Lenac et al., 2006*). We recently found Rhesus CMV (RhCMV) to encode an Fcγ-binding protein in the *Rh05* gene (*RL11* gene family) seemingly more closely related to its HCMV analog (*Kolb et al., 2019*). This is supported by the fact that gpRh05, as HCMV vFcγRs gp34 and gp68, is able to generically antagonize activation of all macaque FcγRs. While it is clear that by targeting the invariant part of the key molecule of the humoral immune response, vFcγRs have the potential to manipulate a multitude of antibody mediated immune functions, their role in vivo has yet to be determined. While the function of HCMV vFcγRs gp34 and gp68 as antagonists of host FcγRs has been established (*Corrales-Aguilar et al., 2014b*), the underlying mechanism(s) had not been addressed yet. In recent years it has been shown that gp68 and gp34 are able to engage in antibody bipolar bridging (ABB) forming ternary complexes consisting of antigen, antibody, and vFcγR (*Corrales-Aguilar et al., 2014a*; *Corrales-Aguilar et al., 2014b*; *Sprague et al., 2008*). Moreover, gp68 has been shown to bind IgG in a 2:1 ratio and has the ability to internalize and translocate IgG to lysosomal compartments, while gp34 has been shown to form predominantly homo-dimeric structures (*Ndjamen et al., 2016*; *Sprague et al., 2008*). However, no

studies have yet been performed in the context of HCMV infection investigating the coincident disposition of gp34 and gp68 at the plasma membrane and their functional interaction during the early and late phase of HCMV replication. Here, we show gp34 and gp68 to antagonize host FcγR activation by distinct but highly cooperative modes of Fcγ targeting, leading to efficient evasion from antibody mediated immune control by division of labor.

## Results

gp34 and gp68 simultaneously bind to distinct regions on IgG. gp68 binding to IgG has been mapped to the CH2–CH3 interdomain region of Fcγ (*Sprague et al., 2008*). Accordingly, in a first experiment we set out to narrow down the contact site of gp34 on IgG utilizing a methodology previously used to characterize HSV-1 gE and HCMV gp68 (*Sprague et al., 2004*; *Sprague et al., 2008*). To this end we infected CV-1 cells with recombinant vaccinia viruses (rVACV) encoding either human FcγRIIA, FcγRI or HCMV vFcγRs gp34 and gp68 (*Sprague et al., 2008*). After metabolic [$^{35}$S]-Met/Cys labeling, Fcγ-binding proteins were precipitated from cell lysates using CNBr-Sepharose coupled with human IgG1-Fc in its wild-type form (wtFc) or as a mutated variant (nbFc) with a scrambled CH2–CH3 interdomain amino acid sequence designed and provided by P. Bjorkman (Caltech, California, USA) (*Sprague et al., 2004*; *Sprague et al., 2008*). Expectedly, gp68 was only able to bind wtFc but not nbFc whereas gp34, comparable to human FcγRI, retained binding to both wtFc and nbFc (*Figure 1A*). While the high affinity FcγRI does not require the CH2–CH3 region to bind to the lower hinge of IgG, FcγRIIA and FcγRIII show lower affinity to monomeric IgG and utilize additional distinct residues for binding to Fcγ, which include the CH2 region adjacent to the lower hinge region of IgG (*Shields et al., 2001*; *Sprague et al., 2008*; *Wines et al., 2000*). Consequently, FcγRII exhibited reduced binding to nbFc (*Figure 1A*). Since gp34 showed intact binding similar to host FcγRI but not FcγRIIA we surmised that gp34 binding involves the hinge region of monomeric IgG. To validate this assumption further gp34 binding to a variant form of hIgG1 with point mutations in the lower hinge region (Leu234Ala and Leu235Ala, named LALA) exhibiting deficient interaction with human FcγRs (*Hessell et al., 2007*) was analyzed (*Figure 1B*). Cell lysates were incubated with either B12 wild-type antibody recognizing human immunodeficiency virus (HIV)-encoded gp120 or a B12-LALA variant. PGS-precipitation of FcγRI/CD64, exclusively recognizing the hinge region on IgG, was almost abrogated by the mutations in the LALA variant. Similarly, gp34 was sensitive to the mutations, albeit to a lower extent than FcγRI/CD64. As expected, the interaction of gp68 with B12 was unaffected by the LALA point mutations. Next, as gp34 and gp68 recognize non-overlapping regions on IgG we set out to test, if both molecules are able to simultaneously associate with IgG. To this end, we generated C-terminal His- or strep-tagged variants of gp34 and gp68 that lack their respective transmembrane and cytosolic domains (soluble gp34 and gp68, or sgp34 and sgp68). These proteins were produced by transfected 293 T cells and secreted into the cell culture supernatant (*Figure 1—figure supplement 1*). Supernatants of sgp34 and sgp68 producing cells were mixed with humanized anti-hCD20 IgG1 (Rituximab, or Rtx) before being precipitated via Ni$^{2+}$-NTA-Sepharose beads. A non-Fcγ-binding point mutant of sgp34 (sgp34mtrp, W65F mutation) was identified and used as a control molecule (*Figure 1—figure supplement 2*). When precipitating His-tagged sgp68 in the presence of monomeric IgG, we observed co-precipitation of sgp34 but no co-precipitation of sgp34mtrp (*Figure 1C*). To further substantiate this observation, we designed an ELISA based setup using the same supernatants as above and performed the assay as depicted schematically in *Figure 1D*. In brief, titrated amounts of strep-tagged sgp34 or sgp34mtrp were immobilized on a 96-well adsorption plate (1° vFcγR) via pre-coated biotin. 1° vFcγRs were then incubated with either Rtx (IgG1) or a Rtx IgA isotype and incubated with His6-tagged sgp68 followed by detection with an anti-His-HRP antibody. Here, we observed dose-dependent binding of sgp68 to titrated amounts of immobilized sgp34. sgp68 could only be detected in the presence of IgG1 but not IgA. Furthermore, sgp68 binding in the presence of IgG1 and sgp34mtrp in this setup does not exceed background levels. Taken together, we conclude that gp34 and gp68 are able to simultaneously bind to IgG with gp34 binding to the upper hinge region and gp68 binding to the CH2–CH3 interface domain on IgG.

gp34 and gp68 exhibit synergistic antagonization of FcγR activation. gp34 (*RL11*) and gp68 (*UL119-118*) have previously been shown to antagonize FcγR activation independently (*Corrales-Aguilar et al., 2014b*). As gp34 and gp68 appear to have a redundant inhibitory function and are

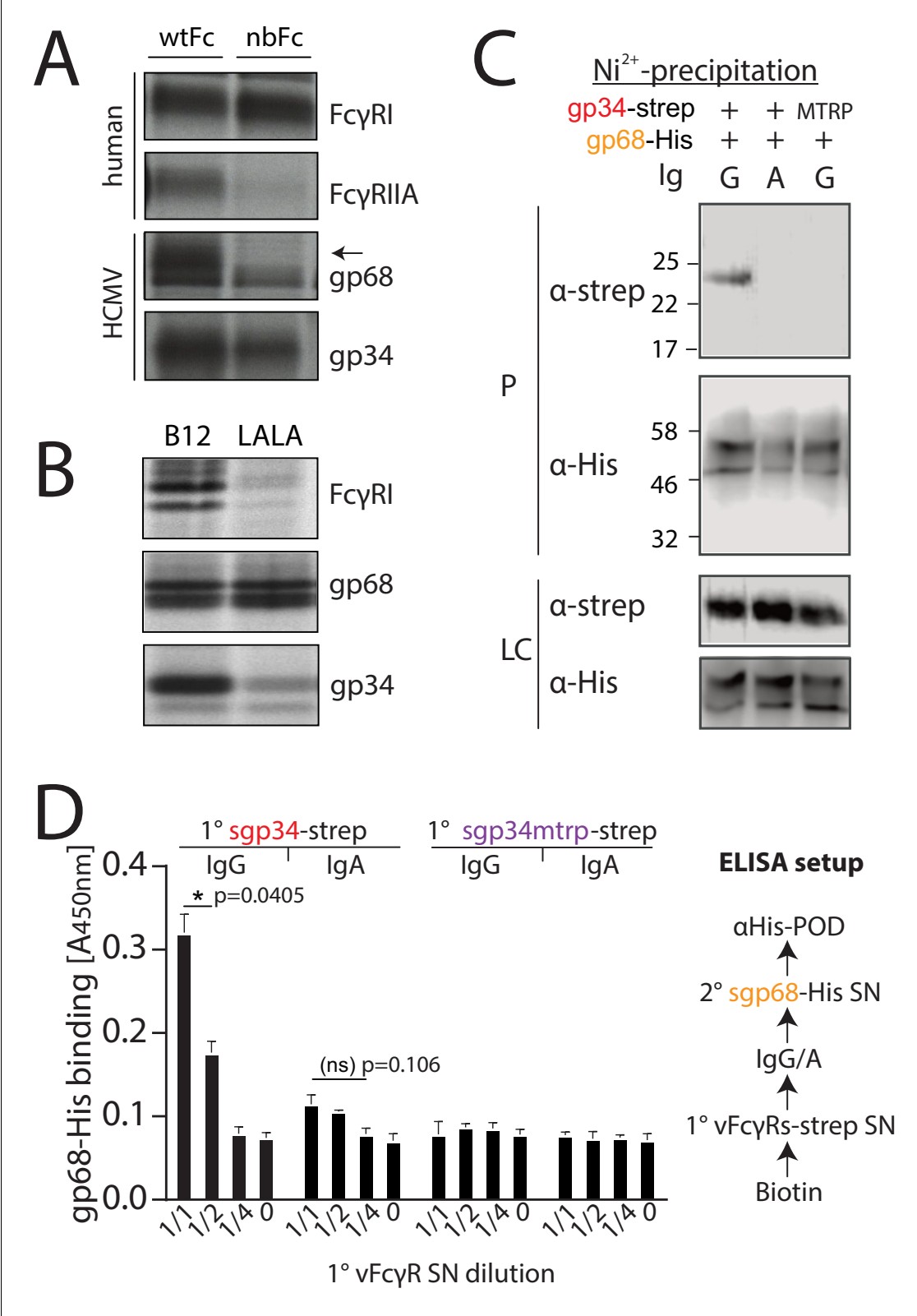

**Figure 1.** gp34 and gp68 simultaneously bind to IgG using distinct epitopes. (**A**) CV-1 cells were infected with rVACVs expressing gp34, gp68, or the host Fc-receptors FcγRIIA and FcγRI at a multiplicity of infection of 4 for 14 hr before metabolic labeling. Proteins were precipitated using either wtFc or nbFc coupled with CNBr-activated Sepharose. Dissociated immune complexes were separated by 10% SDS-PAGE. IP, immunoprecipitation. Shown is one out of two independent experiments. (**B**) CV-I cells were infected and metabolically labeled as above. Lysates were incubated with B12 or B12-

*Figure 1 continued on next page*

*Figure 1 continued*

LALA and IgG was precipitated using PGS. All samples were de-glycosylated using EndoH resulting in double bands for gp68 (*Sprague et al., 2008*). Shown is one out of two independent experiments. (C) Soluble vFcγRs were tested for simultaneous IgG1 (Rtx) binding by Ni-NTA-Sepharose co-precipitation and subsequent immunoblot in the presence of IgG (Rtx). gp34 and gp34mtrp were streptavidin-tagged; gp68 was 6xHis-tagged. All samples were deglycosylated using PNGaseF resulting in double bands for gp68 (*Sprague et al., 2008*). LC = loading control, P = precipitate. Shown is one out of two independent experiments. (D) Soluble vFcγRs as in B were tested for simultaneous IgG1 (Rtx) binding via ELISA as schematically depicted. An anti-CD20 IgA molecule served as a negative control. The 1° sgp34-strep or sgp34mtrp layers were generated by incubation of titrated amounts of supernatants from soluble vFcγR producing cells on coated biotin. Supernatants were diluted in PBS. 2° sgp68-His detection was performed using 1:2 diluted supernatant accordingly. Graph shows averages from two independent experiments performed in technical replicates (*Figure 1—source data 1*). Error bars = SD. Two-way ANOVA.

The online version of this article includes the following source data and figure supplement(s) for figure 1:

**Source data 1.** Relates to *Figure 1D* bar graph.
**Figure supplement 1.** Immunoblot detection of human cytomegalovirus (HCMV) glycoproteins in the supernatant of Hek293T producer cells.
**Figure supplement 2.** Fcγ-binding deficient mutant of gp34.

able to simultaneously bind to IgG we utilized a previously established FcγR activation assay (*Corrales-Aguilar et al., 2013*) to assess their ability to antagonize FcγR activation individually or in combination. In brief, MRC-5 or human foreskin fibroblasts (HFF) were infected with HCMV AD169-pBAC2-derived mutant viruses lacking different combinations of vFcγRs. The other vFcγRs gp95 (*RL12*) and gpRL13 (*RL13*) were deliberately excluded in this study. *RL13* (*Cortese et al., 2012*), a potent inhibitor of HCMV replication (*Stanton et al., 2010*), is notoriously mutated or absent from the HCMV genome in cell culture passages (*Dargan et al., 2010*) but has not yet been described to antagonize FcR activation. *RL12* is one of the most polymorphic genes of HCMV (*Dolan et al., 2004*) with only minor surface expression in the context of AD169 infection (*Corrales-Aguilar et al., 2014b*). Due to remarkably low conservation of *RL12* across HCMV strains, we focused on the well-conserved and previously described vFcγRs gp34 and gp68. Infected cells were incubated with titrated amounts of a HCMV-IgG hyperimmunoglobulin preparation (Cytotect) and co-cultured with BW5147 reporter cells expressing human FcγRIIIA as a chimeric molecule providing the ectodomain of the FcγR fused to the transmembrane domain and cytosolic tail of mouse CD3ξ (*Corrales-Aguilar et al., 2013*). In response to FcγR activation, reporter cells secrete mIL-2 which was then quantified by ELISA (*Corrales-Aguilar et al., 2013*), assay schematically shown in *Figure 2—figure supplement 1*. In line with previously published observations (*Corrales-Aguilar et al., 2014b*) we found that FcγRIIIA activation is antagonized efficiently by a virus co-expressing both gp34 and gp68 while viruses expressing either gp34 or gp68 show only minor or no antagonistic potential when compared with a mutant lacking all vFcγRs (*Figure 2A*). The extent of FcγR antagonization is visualized by area under curve (AUC) comparisons (*Figure 2A* bottom panel). This showed no statistically significant effect for singular vFcγRs, but a strong and significant antagonization by both vFcγRs in combination. To further confirm this key finding we measured degranulation of primary NK cells in response to vFcγR expression to assess antagonization of ADCC. Peripheral Blood Mononuclear Cells (PBMCs) from three donors were incubated with HCMV-infected opsonized human fibroblasts generated as above and CD107a positivity of NK cells was measured after 6 hr of co-culture. In line with the reporter cell assay, primary NK cell activation in response to target cells decorated with anti-HCMV antibodies was antagonized by a virus expressing both gp34 and gp68 with high significance, while the mutant viruses lacking gp34 or gp68 showed drastically lower antagonistic potential (*Figure 2B*). From these observations we conclude that gp34 and gp68 antagonize FcγRs by synergizing modes of action, suggesting cooperation, at least in the absence of gp95 (*RL12*) and gpRL13 (*RL13*).

## HCMV gp34 enhances internalization of immune complexes

As we observed gp34 and gp68 to work in concert to maximize FcγR inhibition, we set out to elucidate the mechanisms by which the vFcγRs achieve this synergistic effect. When comparing the amino acid sequences of the cytosolic domains of gp34 and gp68 it is revealed that both molecules encode distinct sorting motifs (*Figure 3A*; *Atalay et al., 2002*) (source: AD169 pBAC-2 sequence). The cytosolic tail of gp68 harbors an YXXΦ motif seven aa downstream of the transmembrane domain. Such a motif has been described as a marker for lysosomal targeting (*Bonifacino and Traub, 2003*) and is

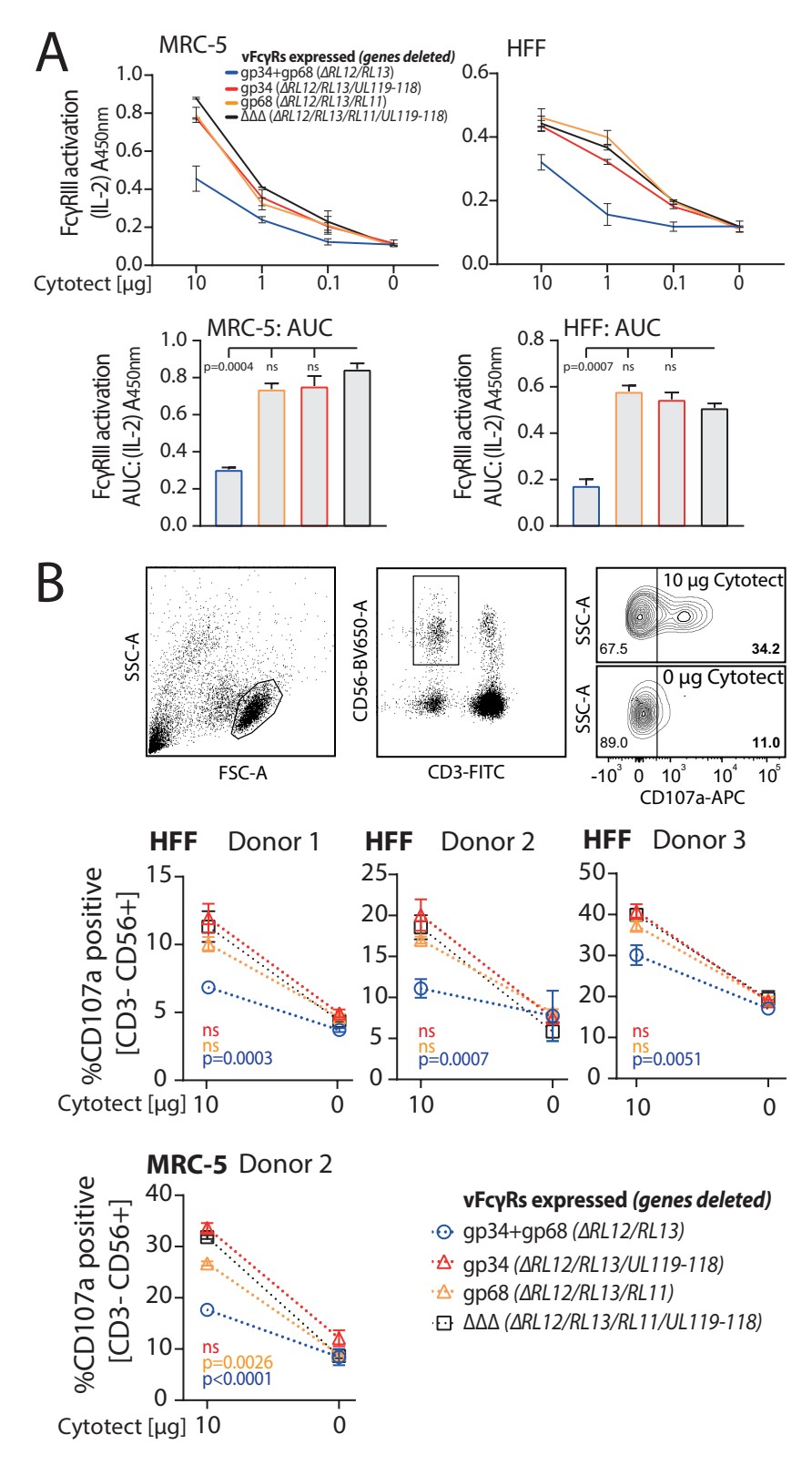

**Figure 2.** gp34 and gp68 synergistically antagonize FcγR activation. (**A**) MRC-5 or human foreskin fibroblasts (HFF) cells were infected with human cytomegalovirus (HCMV) AD169 mutant viruses lacking different combinations of vFcγRs (Multiplicity of Infection [MOI] = 3, for 72 hr) and incubated with titrated amounts of Cytotect. FcγR activation was measured as mIL-2 production of BW5147-FcγRIII reporter cells. Error bars: SD of two independent experiments. Bar graphs show area under curve (AUC) comparison (0.1 cut-off). Error bars: SD. One-way ANOVA. (**B**) Antibody-dependent

*Figure 2 continued on next page*

*Figure 2 continued*

cellular cytotoxicity (ADCC) assay: HFF or MRC-5 cells were infected as in **A** and incubated upon opsonization with Cytotect with PBMCs of three different donors for 6 hr. NK cell CD107a positivity was measured using flow cytometry. Gating strategy shown for one representative experiment of MRC-5 cells infected with HCMV AD169/pBAC2 ΔRL12/UL119-118 incubated with PBMCs from Donor 2. Error bars: SD of two independent experiments performed in technical replicates. Error bars smaller than graph symbols are not shown. Two-way ANOVA compared to ΔΔΔ.

The online version of this article includes the following figure supplement(s) for figure 2:

**Figure supplement 1.** Schematic depiction of the experimental setup of a cell-based FcγR activation assay.

in line with the lysosomal translocation of gp68 shown in a previous study (*Sprague et al., 2008*). In HCMV gB, a non-Fcγ-binding HCMV encoded envelope glycoprotein, the location of two such motifs is further away from the transmembrane domain indicating it being destined for general endocytosis rather than immediate degradation, which is shared by another HCMV vFcγR, gpRL13, found primarily in intracellular compartments as well as HSV-1 gE, forming a heterodimeric gE/gI vFcγR described to internalize IgG and recycle back to the cell surface (*Figure 3—figure supplement 1*; *Bonifacino and Traub, 2003*; *Cortese et al., 2012*; *Ndjamen et al., 2016*; *Sprague et al., 2004*). Conversely, the cytosolic tail of gp34 encodes a di-leucine [D/E]XXX[LL/LI] motif which we find to be unique among surface-resident vFcγRs. Therefore we suspected gp34 and gp68 to possess different dynamics regarding IC internalization. As internalization studies in the context of HCMV infection proved not to be feasible due to lacking gp34- and gp68-specific antibodies for tracing, we aimed to establish a gain-of-function cell-based experimental model. To this end, we characterized the surface dynamics of vFcγR expressing transfected 293 T cells stably expressing a model antigen (hCD20) recognized by Rituximab (RTX). In order to manipulate internalization, we chose to replace the transmembrane and cytosolic domains of gp34 and gp68 with the according sequences from human CD4 (hCD4) as it contains a non-canonical di-leucine-based sorting motif (SQIKRLL) in its C-terminal cytoplasmic tail (CD4-tailed). To be recognized by AP-2 the serine residue upstream of the di-leucine motif in hCD4 needs to be phosphorylated in order to mimic the negatively charged glutamate or aspartate residues of a classical di-leucine motif (*Pitcher et al., 1999*). In a first experiment we ensured equal expression of the transfected constructs utilizing the polycistronic expression of GFP from a pIRES_eGFP expression vector to gate on transfected cells (*Figure 3B*). GFP-positive cells were also detected for Fcγ binding by a surface stain using a PE-TexasRed conjugated human Fcγ fragment (*Figure 3C*). This revealed that upon recombinant expression, surface exposition of Fcγ binding gp68 is drastically increased when CD4-tailed although overall protein expression was comparable and all constructs bear original signal peptides (*Figure 3B*). This can be attributed to the abrogation of its internalization evidenced by the fact that internalization of complete IC was drastically reduced with CD4-tailed gp68 when tracking hCD20/Rtx ICs using pulse-chase flow cytometry (*Figure 3D*). Conversely, while gp34 showed only mild differences in surface exposition when altered in the same way (*Figure 3C*), it proved to rapidly internalize complete Rtx-CD20 IC only with its native cytosolic tail intact (*Figure 3D*). Given that both molecules spontaneously internalize non-immune IgG (*Figure 3—figure supplement 2*), this could hint at different routes of trafficking following internalization. Specifically, as it has been shown that gp68 internalizes IgG-Fc translocating to the lysosomal compartment for degradation (*Ndjamen et al., 2016*), the difference seen for gp34 here suggests a recycling route as described for HSV-1 gE/gI (*Ndjamen et al., 2014*). Further, while we also observe gp34 to more efficiently internalize even non-immune IgG, the still rapid internalization of non-immune IgG by gp68 seems not to translate to the internalization of IC regarding efficiency (*Figure 3—figure supplement 2*), again in line with a previously published observation (*Ndjamen et al., 2016*). As we observed significantly more efficient internalization of IC by gp34wt compared to gp68wt or HSV-1 gEwt (*Figure 3D*, *Figure 3—figure supplement 2*), we conclude that a major task of gp34 compared to gp68 is mediating the efficient internalization of IC, a feature likely linked to its particular di-leucine cytosolic motif.

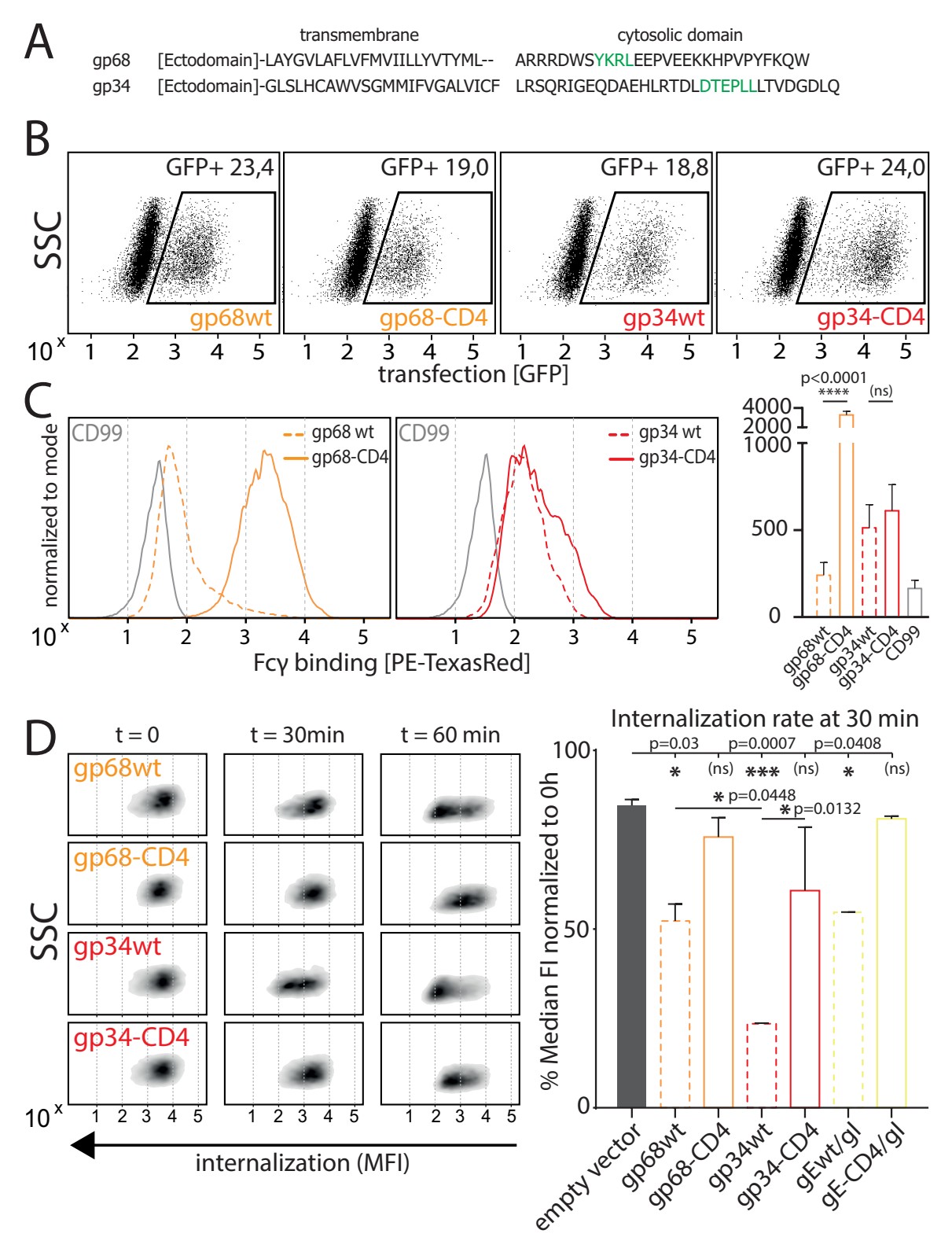

**Figure 3.** Native gp34 efficiently internalizes immune complexes. (**A**) Sequence alignment of human cytomegalovirus (HCMV) AD169 BAC-2 encoded glycoproteins gp34 (*RL11*, P16809) and gp68 (*UL119-118*, P16739). Cellular localization motifs are highlighted in green. (**B**) 293 T cells stably expressing CD20. 293 T-CD20 cells were transfected with indicated constructs encoded by a pIRES_eGFP vector. Transfected cells were gated via GFP expression using flow cytometry. (**C**) GFP positive cells from **B** were measured for surface expression of vFcγRs judged by Fcγ binding. Unaltered native full-length

*Figure 3 continued on next page*

*Figure 3 continued*

molecules (wt) were compared to CD4-tailed variants via a TexasRed-conjugated human Fcγ fragment. CD99 expression from a pIRES_eGFP vector served as a non-Fcγ-binding negative control. Bar graph shows data from three individual experiments. Error bars = SD. One-way ANOVA. (D) Internalization of CD20/Rtx immune complexes in dependence of vFcγR expression was measured by loss of surface signal over time in a pulse-chase approach detecting residual surface complexes via a PE-conjugated mouse-anti-human-IgG antibody. 293 T cells stably expressing CD20 were transfected with the indicated constructs. HSV-1 gE was co-transfected with gI to form a functional heterodimer (*Ndjamen et al., 2014*). Left: Exemplary experiment comparing internalization rates between native vFcγRs (wt) and their respective CD4-tailed variants. Right: Internalization rates 1 hr post pulse. Two independent experiments (*Figure 3—source data 1*). Error bars = SD. One-way ANOVA.

The online version of this article includes the following source data and figure supplement(s) for figure 3:

**Source data 1.** Relates to *Figure 3D* bar graph.
**Figure supplement 1.** Alignment of selected herpesvirus glycoproteins present on the surface of an infected cell.
**Figure supplement 2.** Internalization of CD20/Rtx immune complexes and non-immune IgG in dependence of vFcγR expression.

## FcγRIII binding to opsonized cells is reduced in the presence of gp68 but not gp34

As previously reported (*Atalay et al., 2002*; *Corrales-Aguilar et al., 2014b*) and confirmed in this study, we find that in the context of HCMV infection gp68 is more surface resident compared to gp34 (Figure 6A). Therefore, we next set out to test if the antagonizing effect of membrane-resident gp68 on FcγR activation can be attributed to a block of host FcγR binding to cell surface IC rather than IC internalization. To test this hypothesis, we established a flow cytometry based assay using vFcγR transfected 293 T-CD20 cells opsonized with Rtx and assayed for FcγR binding using soluble His-tagged ectodomains of human FcγRs, which are sequence identical between FcγRs IIIA and IIIB (*Figure 4B*). In this assay we compared gp34 and gp68 regarding their ability to interfere with FcγR binding to cell surface IC. Advantageously, CD4-tailed vFcγRs, besides circumventing confounding effects of internalization on our binding assay, closely mimic the relative surface density of gp34 and gp68 found on the plasma membrane of an HCMV-infected cell judged by human Fcγ binding (*Figure 3C*, *Figure 5A*; *Corrales-Aguilar et al., 2014b*). To ensure equal density of antigen upon recombinant vFcγR expression, CD20 levels were directly measured and found only in the case of CD99 expression to be slightly reduced, which served as a non-Fcγ-binding control molecule (*Figure 4A*). We found gp68 but not gp34 to significantly reduce binding of FcγRIIIA to cell surface immune complexes compared to CD99 (*Figure 4C*). This finding can be explained with certain CH2–CH3 region residues of IgG playing a subordinate, yet significant role in FcγRIII binding to IgG (*Shields et al., 2001*). Additionally, we found the effect of gp68 to be dependent on the accessibility of the CH2–CH3 interface region on IgG as pre-incubation with Protein G prevented the inhibition by gp68 and fully restored FcγRIIIA binding to Rtx (*Figure 4C*). These observations further narrow down the binding region of gp68 to involve the above-mentioned residues as opposed to Protein G which has been shown to not interact with these residues (*Sauer-Eriksson et al., 1995*). Finally, this conclusion is further substantiated by our observation that FcγRI, which binds to IgG independently of the above-mentioned residues in the CH2–CH3 interdomain region (*Shields et al., 2001*), is not blocked by gp68 or gp34 (*Figure 4—figure supplement 1*).

## Human FcγR binding to opsonized HCMV-infected cells is reduced by viral gp68

In order to test if our previous findings regarding the block of human FcγRIII binding to immune complexes in the presence of gp68 translate into a loss-of-function approach, MRC-5 cells were infected with AD169/pBAC2-derived HCMV mutants as described above. Uniformity of HCMV antigen expression and differential surface Fcγ binding indicating vFcγR expression between the different virus mutants was assured using a F(ab')$_2$ preparation of purified human IgG containing HCMV-specific IgG, Cytotect, or a conjugated human Fcγ fragment (human Fcγ-Texas Red, TR), respectively (*Figure 5A*). FcγR binding to opsonized-infected cells was detected by flow cytometry as described above and schematically depicted in *Figure 5B*. Expectedly, as we observed stronger Fcγ binding to cells infected with gp68 proficient viruses compared to a gp34 expressing virus (*Figure 5A*), we also found more total IgG bound to the surface of cells infected with a gp68 expressing HCMV virus compared to a gp34 expressing virus (*Figure 5C*, upper panel). Along these lines, we find FcγRIII to bind

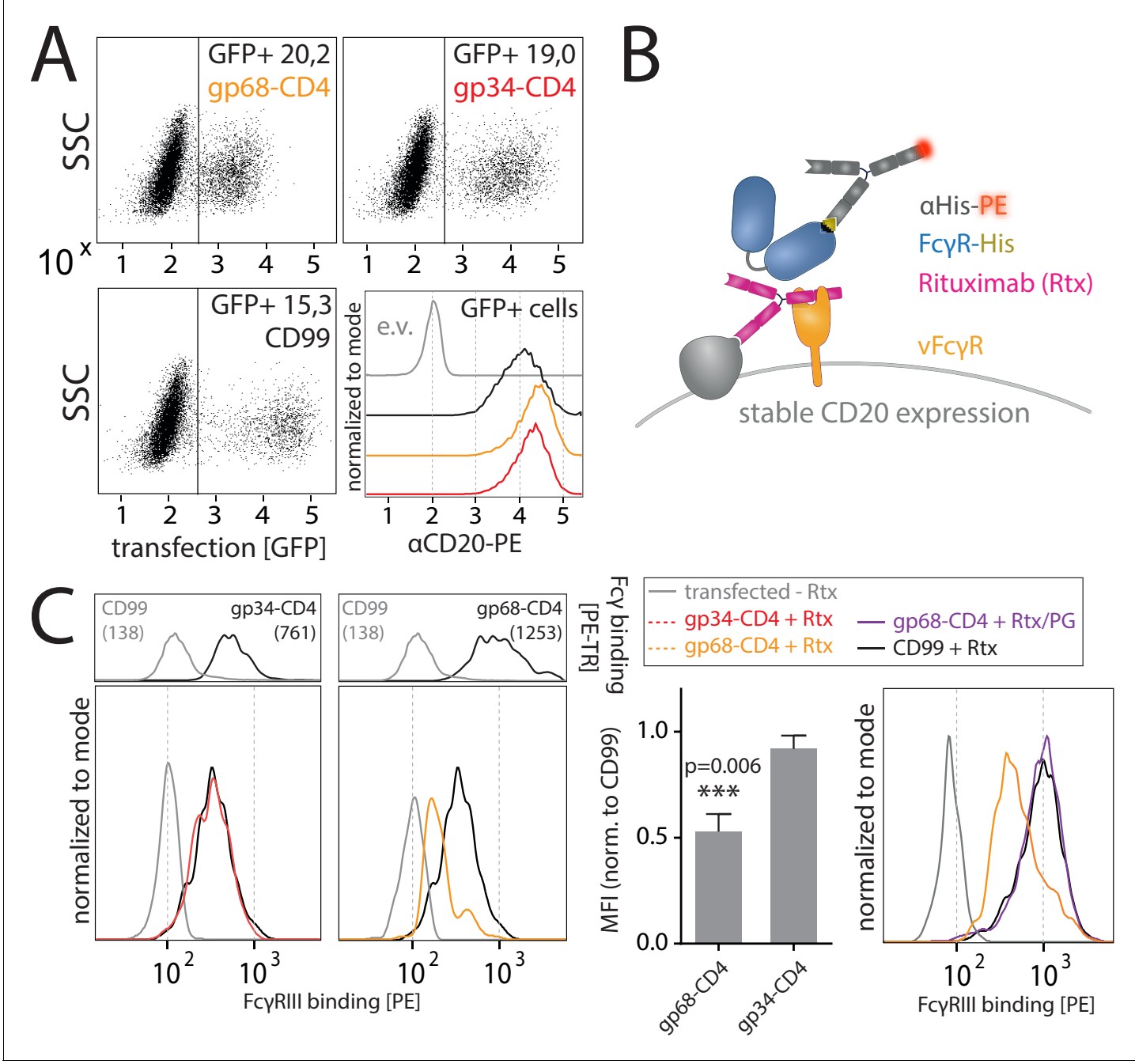

**Figure 4.** gp68 blocks binding of human FcγRIII to immune complexes. (**A**) 293 T-CD20 cells were transfected with indicated constructs encoded on a pIRES_eGFP vector. CD99 served as a non-Fcγ-binding control. Gating strategy and CD20 expression in dependence of vFcγR or CD99 co-expression. CD20 was detected using a PE-labeled αCD20 antibody. (**B**) Schematic depiction of the experimental flow cytometry setup to measure FcγR binding to cell surface IC. (**C**) Expression of vFcγRs or CD99 differentially modulates binding of FcγRIII to CD20/Rtx complexes. Total IgG binding by vFcγRs is demonstrated by Fc binding (PE-TexasRed-conjugated Fcγ fragment) compared to CD99 (upper panels). MFI are indicated in brackets. Pre-incubation of Rtx with Protein G counteracts block of FcγRIII binding mediated by gp68. Bar graph: Comparing the blocking effect of vFcγRs over four independent experiments normalized to CD99 (*Figure 4—source data 1*). Error bars = SD. Two-tailed t-test.

The online version of this article includes the following source data and figure supplement(s) for figure 4:

**Source data 1.** Relates to *Figure 4C* bar graph.

**Figure supplement 1.** gp68 is not able to block FcγRI binding to IC.

**Figure supplement 1—source data 1.** Relates to *Figure 4—figure supplement 1* bar graph.

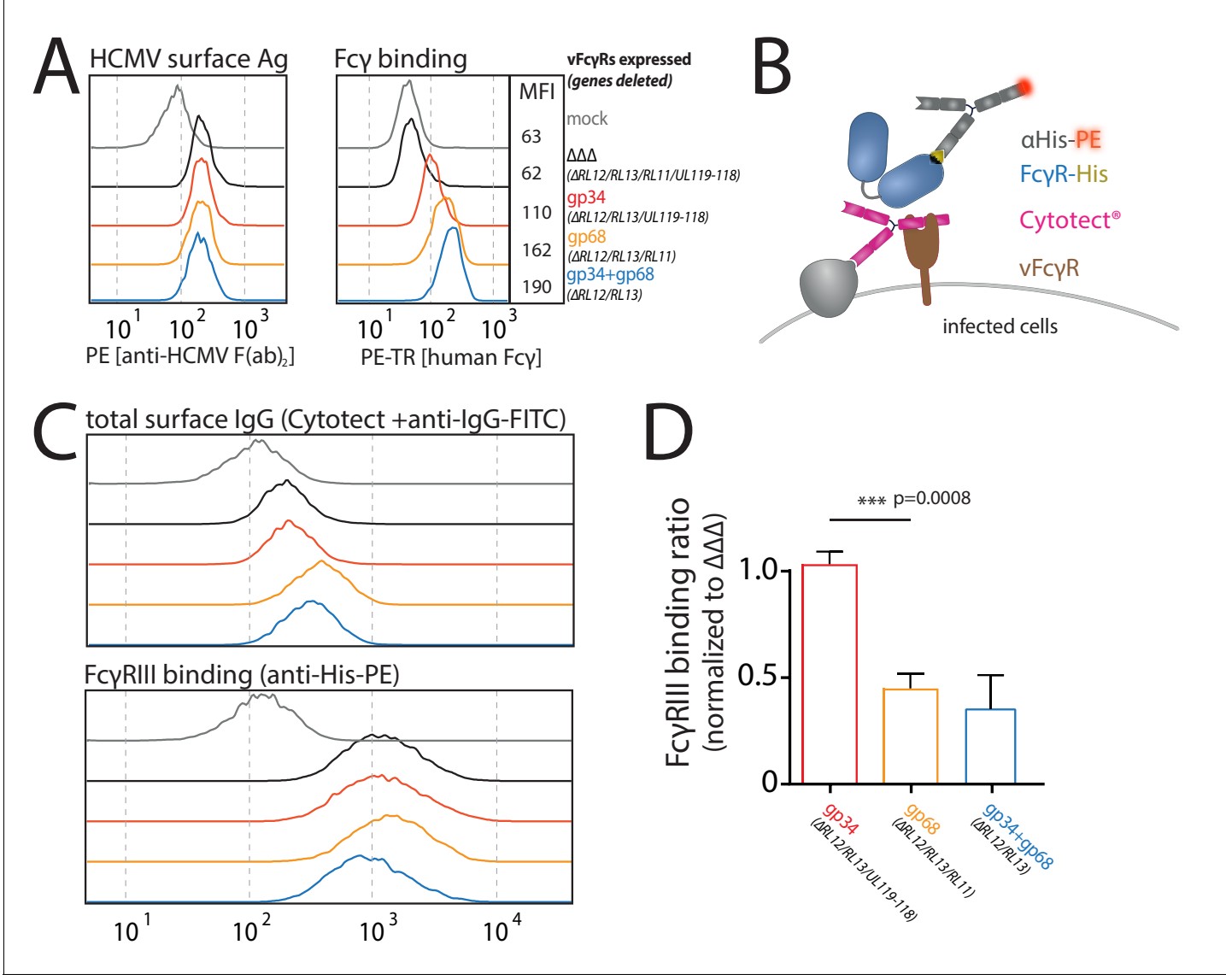

**Figure 5.** Human cytomegalovirus (HCMV) gp68, but not gp34 blocks binding of human FcγRIII to opsonized HCMV-infected cells. MRC-5 cells were infected with mutant HCMV Ad169 viruses lacking different vFcγR encoding genes at MOI = 2 and measured 72dpi via flow cytometry. ΔΔΔ = ΔRL11/Δ RL12/ΔUL119-118. ΔUL119 = ΔUL119-118. (A) Left: Infected cells were measured for surface expression of antigens using Cytotect derived F(ab')₂ fragments detected via polyclonal rabbit anti-human-IgG-FITC. Right: The same cells were measured for surface binding of human Fcγ using a TexasRed conjugated human Fcγ fragment. (B) Schematic depiction of the experimental setup to measure human FcγR binding to immune complexes in dependence of vFcγR deletion. (C) Representative experiment showing the binding of soluble ectodomains of His-tagged human FcγRIIIA to infected cells opsonized with Cytotect and the total surface bound IgG. FcγR binding was detected via αHis-PE staining. Effects on FcγRIIIA binding are obscured by total surface IgG binding varying between viruses. (D) Three independent experiments testing FcγRIII binding to infected cells incubated with Cytotect are shown in one graph. FcγRIII binding to infected cells pre-incubated with Cytotect was calculated as MFI ratio between FcγR-binding and surface-IgG to account for differences in surface attachment of human antibodies in the presence of different combinations of vFcγRs. FcγRIIIA binding was normalized to ΔΔΔ for each individual experiment (*Figure 5—source data 1*). Error bars = SD, one-way ANOVA.

The online version of this article includes the following source data and figure supplement(s) for figure 5:

**Source data 1.** Relates to *Figure 5D* bar graph.

**Figure supplement 1.** Gp68, but not gp34, binds IgG simultaneously to FcγRIII.

**Figure supplement 2.** Human cytomegalovirus (HCMV) gp68 reduces binding of human FcγRII ectodomains to opsonized-infected cells.

gp68-bound non-immune IgG (*Figure 5—figure supplement 1*), albeit at a markedly lower rate compared to immune IgG (*Figure 4*) in line with FcγRs naturally showing higher affinity toward antigen-bound IgG (*Bruhns et al., 2009*). As the Cytotect formulation contains an abundance of non-immune IgG, binding of host FcγR ectodomains as shown exemplarily in *Figure 5C* (lower panel) was normalized to levels of total surface bound IgG within each experiment (*Figure 5C*, upper panel). Evaluating the results from three independent experiments performed with independent virus preparations confirmed a strong relative reduction in FcγRIIIA binding to opsonized HCMV-infected cells in the presence of gp68, but not gp34 (*Figure 5D*). This finding is in line with our previous gain-of-function experiments (*Figure 4C*) again showing an approximate 60% reduction in FcγRIII binding. While not further elaborated on in this study, we also measured binding of FcγRs I, IIA, and IIB/C in the same setup and observed that binding of FcγRI was only slightly reduced in the presence of either vFcγR, but clearly reduced in the presence of both molecules. Conversely, gp68 showed a similar effect on FcγRs IIA and IIB/C as observed for FcγRIII (*Figure 5—figure supplement 2*). This indicates a comparable antagonistic mechanism of gp68 on FcγRs IIA and IIB/C but shows again FcγRI to be more resistant to gp68.

## Cooperative antagonization of FcγR activation is achieved by combining gp68-mediated block of FcγR binding and subsequent gp34 internalization

While we delineated distinct mechanisms by which gp34 or gp68 are able to counteract FcγRII/III activation, we did not observe efficient antagonization of FcγR function by gp34 or gp68 individually in the context of HCMV deletion mutant infection (*Figure 2*). Therefore, we next wanted to test the antagonistic potential of gp68 or gp34 individually in the absence of non-immune IgG and in a gain-of-function setting. To this end, we conducted an FcγR activation assay with Hela cells co-transfected with Her2 as a model antigen and the indicated vFcγRs followed by incubation with titrated amounts of anti-Her2 IgG1 mAb (herceptin, Hc) (*Figure 6A*). Using this approach, gp68 or gp34 individually confirmed to antagonize FcγRIII activation. However, the more membrane resident gp68-CD4 showed a markedly stronger antagonistic effect compared to gp68wt, indicating a more prominent membrane-residence of gp68 to be beneficial regarding evasion from FcγR recognition despite its lower capacity to internalize ICs (*Figure 3D*). Conversely, when comparing unaltered gp34 to its less internalized gp34-CD4 variant we observed an opposite effect indicating internalization to be a major condition of gp34 driven antagonization of FcγR activation. Next, in an approach mimicking HCMV immune sera we tested if the addition of non-antigen-specific IgG impairs the antagonization of FcγR activation by gp34 or gp68. To this end, we added 10 µg/ml TNFα-specific Infliximab (Ifx) IgG1 given concomitantly with the reporter cells and graded amounts of Hc IgG1 (*Figure 6B*). This showed that indeed the presence of an excess of non-immune IgG interferes with both gp68 and gp34 antagonization in a dose-dependent manner. As ultrapure monomeric IgG does not activate the reporter cells on its own, shown by the samples that were treated only with Ifx but not Hc, this effect can be attributed to displacement of immune IgG from the vFcγRs resulting in restoration of FcγRIII activation. Synergism between gp34 and gp68 was optimal at a 100:1 ratio of non-immune Ifx over Hc (*Figure 6B*, right panel) indicating that vFcγRs are particularly adapted to work in the presence of excess non-immune IgG, supporting our previous observation with human hyperimmunoglobulin Cytotect (see *Figure 2*). On the other hand, host FcγRs have been shown to bind IgG immune complexes with a higher affinity compared to monomeric IgG (*Bruhns et al., 2009*), limiting the efficacy of monomeric IgG in attenuating FcγR activation. Finally, we addressed cooperative antagonization by gp34 and gp68 in a loss-of-function approach in the context of HCMV infection. When using the humanized anti-HCMV mAb MSL-109 targeting HCMV gH, we found that low gH levels expressed on the surface of HCMV-AD169-infected cell are insufficient to detect FcγR triggering (*Figure 6—figure supplement 1*). To overcome this experimental problem, we generated Her2-expressing BJ fibroblasts exhibiting high levels of Her2 on the cell surface. Infected Her2 BJ fibroblasts were opsonized with titrated amounts of Herceptin before cells were analyzed by FcγRIIIA reporter cells (*Figure 6C*). Although not statistically significant, this approach clearly demonstrated that compared to our observations with hyperimmunoglobulin Cytotect (see *Figure 2*), natively expressed gp34 as well as gp68 are able to antagonize FcγRIIIA activation individually. When both vFcγRs were expressed by HCMV-infected cells, a markedly stronger antagonistic effect was seen (p=0.0252).

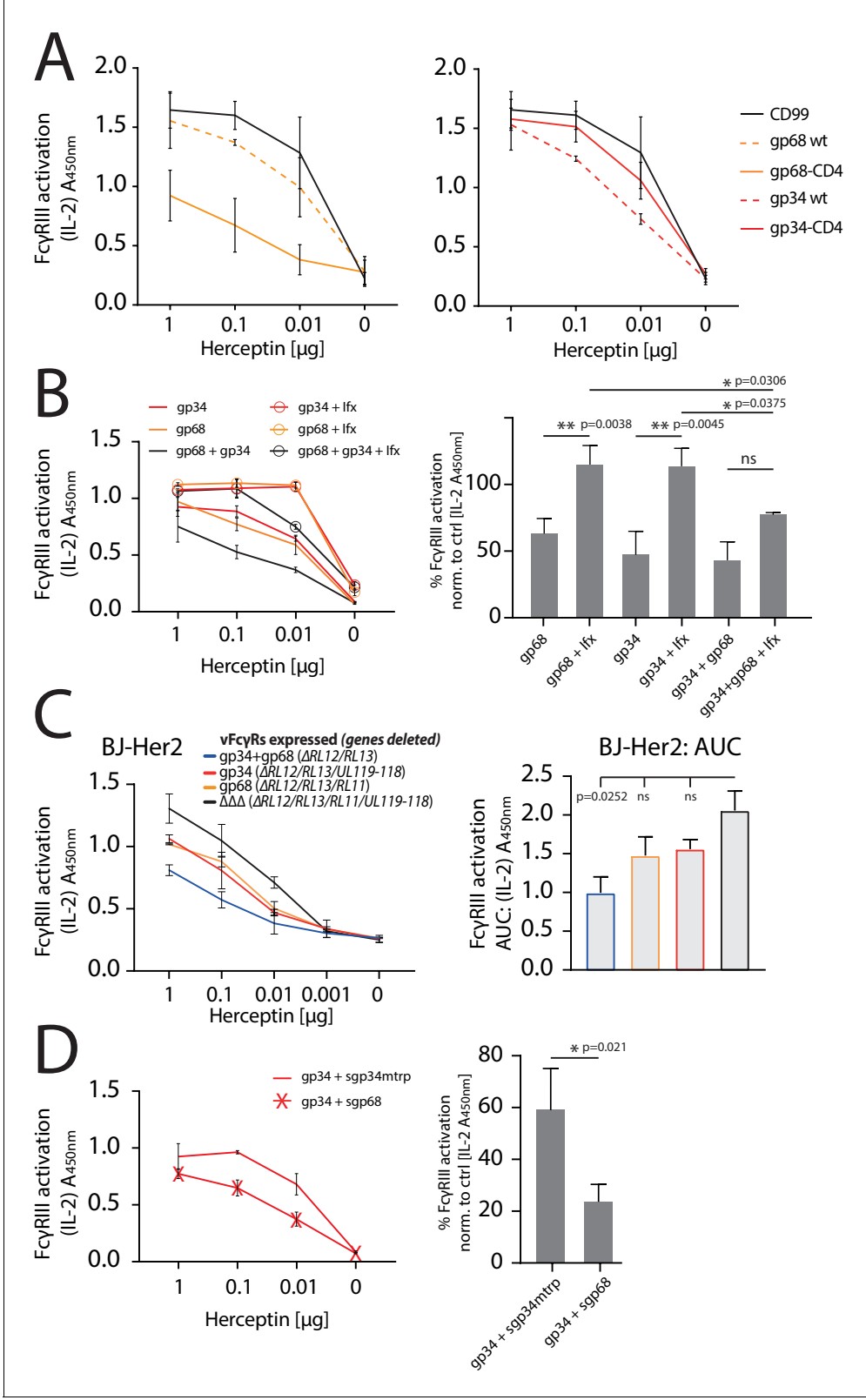

**Figure 6.** gp34 and gp68 synergistically antagonize. FcγRIIIA in the absence and presence of non-immune IgG and show cooperativity. (**A**) Increased membrane-residence enhances antagonistic potential of gp68 and reduces antagonistic potential of gp34. Hela cells were co-transfected as indicated in combination with Her2 antigen and incubated with titrated amounts of Herceptin followed by FcγRIII activation assessment. Three independent experiments performed in technical replicates. Error bars = SD. (**B**) Non-immune IgG counteracts antagonistic potential of vFcγRs. Hela cells were co-

*Figure 6 continued on next page*

*Figure 6 continued*

transfected as indicated in combination with Her2 antigen and incubated with titrated amounts of Herceptin (µg per 100 µl) followed by FcγRIII activation assessment. FcγR activation was assessed in the absence or presence of excess of non-immune IgG by addition of 1 µg per 100 µl Infliximab (Ifx). Hc, Ifx, and reporter cells were added concomitantly. Titrations show one exemplary experiment. Error bars = SD. Bar graphs show combined averages from three independent experiments performed in technical replicates at 0.01 µg Hc normalized to CD99 control (*Figure 6—source data 1*). Error bars = SD. Two-way ANOVA. (C) gp34 and gp68 antagonize FcγRIIIA activation individually in the context of viral infection. BJ-Her2 cells were infected with the indicated human cytomegalovirus (HCMV) AD169 deletion mutants (MOI = 2). 72 hr post infection, cells were opsonized with titrated amounts of Herceptin and incubated with FcγRIIIA reporter cells. mIL-2 expression was quantified via anti-mIL-2 sandwich ELISA. Two independent experiments performed in technical replicates. Error bars = SD. Bar graph shows area under curve (AUC) comparison (*Figure 6—source data 2*). Error bars = SD. One-way ANOVA. (D) gp68 cooperatively enhances antagonistic potential of gp34. Hela cells were co-transfected with gp34 and Her2 antigen and incubated with titrated amounts of Herceptin followed by FcγRIII activation assessment. gp34 expressing cells were supplemented with sgp68 or sgp34mtrp from soluble vFcγR producer cells at a 1:4 dilution. Soluble vFcγRs and reporter cells were incubated concomitantly after removal of pre-incubation with Hc to avoid saturation of sgp68 with unbound Hc. Titrations show one exemplary experiment. Error bars = SD. Bar graphs show combined averages from three independent experiments performed in technical replicates at 0.01 µg Hc normalized to CD99 control (*Figure 6—source data 3*). Error bars = SD. Unpaired t-test.

The online version of this article includes the following source data and figure supplement(s) for figure 6:

**Source data 1.** Relates to *Figure 6B* bar graph.
**Source data 2.** Relates to *Figure 6C* bar graph.
**Source data 3.** Relates to *Figure 6D* bar graph.
**Figure supplement 1.** Human cytomegalovirus (HCMV) gH surface expression on HCMV-infected cells.

Following the reveal of synergistic modes of action, we next explored cooperativity between gp34 and gp68 in a setup of reduced complexity. Specifically, in order to elaborate on the cooperativity of gp34 and gp68 regarding the here highlighted main mechanisms of internalization (gp34) and host FcγR-binding blockade (gp68) we chose to co-express gp34 and Her2 antigen while adding soluble gp68 (sgp68, *Figure 1—figure supplement 1*), or a soluble functional deficient gp34 point mutant as a control (sgp34mtrp, *Figure 1—figure supplement 1* and *Figure 1—figure supplement 2*), together with the reporter cells after the removal of unbound Hc. Here we observe that the ectodomain of gp68 is sufficient to enhance the antagonistic effect of gp34 (*Figure 6C*).

In summary, we conclude that (i) gp34 is designed for internalization of IC while gp68 blocks FcγR binding to IC; (ii) gp34 and gp68 are able to antagonize FcγR activation individually when faced with titrated amounts of immune IgG, but non-immune IgG interferes with this inhibition; (iii) gp34 and gp68 show cooperativity in attenuating FcγR activation particularly under conditions of high excess of non-immune IgG.

## Discussion

Here we delineate for the first time synergistic modes of action involving two independent viral factors to efficiently achieve evasion from antibody-mediated immune cell surveillance (*Figure 7*). The novelty of these mechanisms acting cooperatively on the surface of an infected cell has major implications for our perception of how viruses have co-evolved multifactorial strategies to compete in a molecular arms race with the host at the junction between adaptive (humoral) and innate immunity. In particular, this knowledge refines our understanding of how HCMV can efficiently evade from a fully developed, broadly polyclonal and affinity-matured IgG response using its large coding capacity to ensure its reactivation, shedding, spread, and eventually transmission.

Recognizing the conserved Fc part of IgG and exhibiting a very similar FcγR antagonization profile in reductionistic gain-of-function approaches (*Corrales-Aguilar et al., 2014b*), the two membrane-resident glycoproteins gp34 (*RL11*) and gp68 (*UL119-118*) appeared at first glance merely redundant. This left the puzzling question as to why HCMV (and cytomegaloviruses in general) have developed a whole arsenal of antagonists, unlike other viral and microbial pathogens with only singular Fc-binding glycoproteins (*Berry et al., 2020*; *Corrales-Aguilar et al., 2014a*; *Falugi et al., 2013*). However, when carefully examining their effect in a loss-of-function approach based on opsonized cells infected with a complete set of targeted HCMV gene deletion mutants, we convinced ourselves that gp34 and gp68 actually depend on one another to efficiently antagonize NK cell degranulation (*Figure 2*). This is explained by a number of individual features of gp34 and gp68 that only in conjunction can amount to antagonistic efficiency.

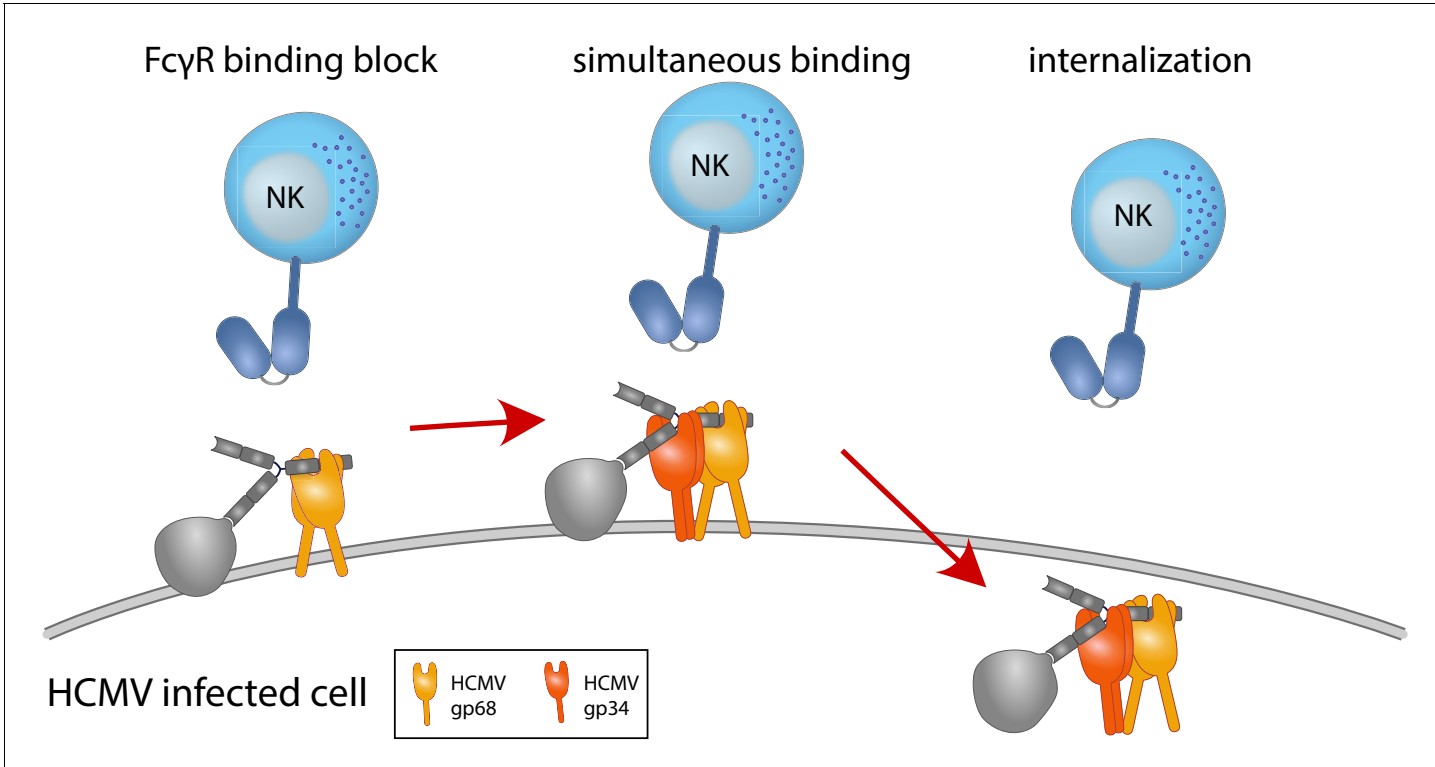

**Figure 7.** Graphical summary. NK cells elicit a powerful antibody-mediated antiviral response through antibody-dependent cellular cytotoxicity (ADCC). gp68 (ochroid) binds IgG in a 2:1 ratio reducing, but not abolishing accessibility of immune complexes to FcγRIII[+] (dark blue) immune effector cells such as NK cells. gp34 (red, natively forming a dimer [*Sprague et al., 2008*]) effectively internalizes immune complexes making them unavailable to surveilling FcγRIII[+] effector cells but cannot compete with FcγRIII for a similar binding region on IgG. Supported by functional evaluation, we propose a model in which g34 and gp68 work in cooperation to achieve efficient antagonization of antibody-mediated effector mechanisms.

## gp34 efficiently internalizes IC

While we find both, gp34 and gp68, able to internalize monomeric IgG as well as IC (*Figure 3*, *Figure 3—figure supplement 2*), gp34 consistently shows more rapid internalization compared to gp68 (*Figure 3*). Consequently exchange of the cytosolic domain of gp34 leads to strongly reduced internalization and subsequently to a reduction in FcγR antagonization (*Figure 6*). Looking closely at the cytosolic domains of all identified cytomegalovirus vFcγRs reveals a unique [D/E]XXX[LL/LI] cytosolic motif present in gp34. Conversely, the HCMV FcγRs gp68, gp95, and gpRL13 all share only a cytosolic YXXΦ motif, as the recently identified RhCMV encoded vFcγR gpRH05 (*RL11* gene family) and its homologs which are conserved in Old World monkey CMV species (*Kolb et al., 2019*). Further comparing the cytosolic sorting motifs between other membrane-resident glycoproteins of HCMV it seems that the YXXΦ motif of gp68 fulfills a more general purpose of limiting the overall surface exposure of concerned HCMV antigens (examples listed in *Figure 3—figure supplement 1*; *Corrales-Aguilar et al., 2014b*; *Ndjamen et al., 2016*). While the YXXΦ motif of gp68 lies within seven aa distance to the transmembrane domain, in line with it being translocated to lysosomal compartments (*Bonifacino and Traub, 2003*; *Ndjamen et al., 2016*), the YXXΦ motif is additionally linked to specific targeting functions. For example, HSV-1 gE has been shown to bind in a pH-dependent manner not observed for any other HCMV encoded vFcγR (*Sprague et al., 2004*; *Sprague et al., 2008*) which fits to it being destined for recycling rather than degradation (*Ndjamen et al., 2014*). This is in line with its YXXΦ motif being more distant to its transmembrane domain within around seven amino acids (*Bonifacino and Traub, 2003*; *Figure 3—figure supplement 1*). Similar membrane-distal YXXΦ motifs also exist in the cytosolic domains of HCMV encoded gpRL13 and gB, but not the model antigen Her2 which we explored in *Figure 6C*, perhaps explaining the limited synergistic inhibition that was observed. These findings highlight the di-leucine motif

found in gp34 to be unique among other surface-resident glycoproteins expressed by HCMV and all other vFcγRs known in CMV family members including RhCMV and MCMV (*Kolb et al., 2019*; *Thäle et al., 1994*). It remains to be explored as to what the further consequences of this seemingly unique internalization route is. However, as we also find gp34 to be incorporated into the virion (*Reinhard, 2010*) it is tempting to speculate that the potent internalization of IC via gp34 has an additional role besides evasion from antibody-mediated attack of an infected cell.

## gp68 blocks binding of FcγRs II and III to IC

Although we find gp34 to efficiently internalize ICs from the cell surface, it does not manipulate host FcγR binding to ICs. This is remarkable given our data showing that gp34 binds to the hinge region of monomeric IgG, but implying that gp34 is not able to directly compete with host FcγRs for binding to IC on the plasma membrane when co-expressed with a target antigen. Conversely, gp68 binding to the CH2–CH3 interface domain also occupies a region on Fcγ that includes residues involved in FcγRII and FcγRIII binding to Fcγ (*Shields et al., 2001*; *Sprague et al., 2008*). This could explain the ability of gp68 to efficiently limit binding of FcγRs II and III to cell surface IC without directly competing for a shared binding region at the hinge region (*Figure 1*, *Figures 4* and *5*). The subordinate role of the above-mentioned residues on IgG regarding FcγRIII binding further ties into the observation that the blocking effect of gp68 is not total but more in line with the reported reduction of FcγRIII binding to IgG being approximately 40–70% when these residues are manipulated (*Shields et al., 2001*). Conversely, gp68 not showing a similar efficiency in blocking FcγRI binding to IC further supports this conclusion as FcγRI binding does not require the above mentioned CH2–CH3 region residues (*Figure 3—figure supplement 1*; *Shields et al., 2001*).

## gp34 and gp68 cooperatively antagonize FcγR activation

Taken together, we conclude that neither gp68 nor gp34 individually are able to efficiently antagonize FcγR activation by IgG-opsonized viral antigens in the physiological context of HCMV infection (*Figure 2*), that is, in the presence of non-immune IgG mitigating their inhibitory potential (*Figure 6B* and *Figure 6C*). Our data provide evidence that gp68 rather functions to increase IC accessibility to gp34 leading to efficient internalization of IC, presumably by a repeating process of gp34 recycling. This is supported by our observation that the ectodomain of gp68 is sufficient to increase the antagonistic efficiency of gp34 regarding FcγR activation (*Figure 6*). Further, gp34 and gp68 are co-expressed at the cell surface throughout the protracted HCMV replication cycle and show simultaneous Fcγ binding (*Figure 1*). Our findings imply that gp68 binding, as it does not use the same region as host FcγRs, evolved to ensure a consistent and evolutionary less vulnerable way of shifting Fcγ accessibility in favor of gp34. The idea that gp68 has not evolved to directly compete with host receptors for IgG binding is also supported by the finding demonstrating gp68 to possess a lower affinity to Fcγ when binding in a 2:1 ratio compared to host FcγRIIIA ($K_{D1}$:470 nM and $K_{D2}$:1.600 nM vs 700 nM) (*Li et al., 2007*; *Sprague et al., 2008*). The concept of gp68 driven accessibility shift also does not require gp68 to completely block FcγR binding to IC as subsequent gp34-driven internalization is a continuous and fast process.

## Beyond NK cells

In this study we mainly focused on the mechanisms by which gp34 and gp68 cooperate to antagonize CD16/FcγRIII as the primary receptor on NK cells associated with ADCC, one of the most powerful mechanisms of antibody-mediated virus control. However, it is already known that HCMV vFcγRs also antagonize activation of FcγRIIA (CD32) and FcγRI (CD64) (*Corrales-Aguilar et al., 2014b*) and we also consistently observed antagonization of the only inhibitory FcγR, FcγRIIB (unpublished observation). Similarly, we could recently show antagonization of all canonical Rhesus FcγRs I, IIA, IIB, and III by the RhCMV encoded vFcγR gpRH05 (*RL11* gene family) (*Kolb et al., 2019*). Taken together with the data shown in this study that the underlying mechanism seems to be related between FcγRs IIA/B and III, we also speculate that more Fcγ-binding host factors might be antagonized by similar mechanisms involving vFcγRs. In support of this view, besides gp34 and gp68, the HCMV *RL11* gene family encodes additional vFcγRs. *RL12* (gp95) and *RL13* (gpRL13) likely are part of a larger arsenal of antibody targeting immunoevasins (*Corrales-Aguilar et al., 2014a*; *Cortese et al., 2012*). The fact that the interaction of gp34 and gp68 was analyzed in the absence

of gp95 (RL12) and gpRL13 (RL13) is a major limitation of our study. Synergies between other or even all four vFcγRs are also conceivable and subject to further investigation. Altogether, our findings analyze the first mechanistic details of HCMV evasion from antibody-mediated control utilizing its vFcγR toolset. This deeper insight into the inner workings of such a process has consequences for the future evaluation and optimization of antibody-based treatment strategies targeting HCMV disease. In particular, our data will support the development of targeted intervention strategies that neutralize the function of gp34 and gp68, to increase the efficiency of IVIg treatment in the future.

# Materials and methods

## Key resources table

| Reagent type (species) or resource | Designation | Source or reference | Identifiers | Additional information |
|---|---|---|---|---|
| Gene (HCMV) | UL119-118 | This paper | MN900952.1 | |
| Gene (HCMV) | RL11 | This paper | MN900952.1 | |
| Gene (Homo sapiens) | CD4 | This paper | BT019791.1 | |
| Gene (HSV-1) | US8 | This paper | MN136524.1 | |
| Gene (HSV-1) | US7 | This paper | MN136524.1 | |
| Strain, strain background E. coli | NEB5-alpha | NEB | C2987 | Made chemically competent for cloning via CaCl$_2$ |
| Strain, strain background (HCMV) | AD169-BAC2 | doi:10.1016/j.celrep.2020 | MN900952.1 | |
| Genetic reagent Mus musculus | BW5147 FcγR-reporter cells | doi:10.1016/j.jim.2012.09.006 | | |
| Genetic reagent (H. sapiens) | Hek-CD20 | Kindly provided by Irvin Chen, UCLA | | Lentiviral transduction |
| Genetic reagent (H. sapiens) | BJ-Her2 | This paper | | Lentiviral transduction as in doi: 10.1128/JVI.01923-10 |
| Cell line (H. sapiens) | Hela | ATCC | CCL-2 | |
| Cell line (H. sapiens) | MRC-5 | ECACC | 05090501 | |
| Cell line (H. sapiens) | HFF Human foreskin fibroblasts | Kindly provided by Dieter Neumann-Haefelin and Valeria Kapper-Falcone, Institute of Virology, Freiburg, Freiburg, Germany | HF-99/7 | |
| Cell line (H. sapiens) | BJ-5ta | ATCC | CRL-4001 | |
| Cell line (H. sapiens) | 293T-CD20 | Kindly provided by Irvin Chen, UCLA, USA | | |
| Cell line (M. musculus) | BW5147 | Kindly provided by Ofer Mandelboim, Hadassah Hospital, Jerusalem, Israel | | FcR-expressing cell lines as in *Corrales-Aguilar et al., 2013* |

*Continued on next page*

*Continued*

| Reagent type (species) or resource | Designation | Source or reference | Identifiers | Additional information |
|---|---|---|---|---|
| Cell line (*H. sapiens*) | PBMC | Primary human cells | | Primary cells isolated from human donors |
| Transfected construct (*H. sapiens*) | Her2/Erbb2 | gBlock by IDT | NM_004448 | Construct to generate stably expressing BJ cells |
| Transfected construct (*H. sapiens*) | gp68 | gBlock by IDT | UL119-118 of MN900952.1 | Cloning via added flanking Nhe1 and BamH1 restriction sites |
| Transfected construct (*H. sapiens*) | gp34 | gBlock by IDT | RL11 of MN900952.1 | Cloning via added flanking Nhe1 and BamH1 restriction sites |
| Transfected construct (*H. sapiens*) | gE | gBlock by IDT | US8 of MN136524.1 | Cloning via added flanking Nhe1 and BamH1 restriction sites |
| Transfected construct (*H. sapiens*) | gI | gBlock by IDT | US7 of MN136524.1 | Cloning via added flanking Nhe1 and BamH1 restriction sites |
| Transfected construct (*H. sapiens*) | gp68-CD4 | gBlock by IDT | UL119-118 of MN900952.1 fused to human CD4 TM and cytosolic tail BT019791.1 | Cloning via added flanking Nhe1 and BamH1 restriction sites |
| Transfected construct (*H. sapiens*) | gp34-CD4 | gBlock by IDT | RL11 of MN900952.1 fused to human CD4 TM and cytosolic tail BT019791.1 | Cloning via added flanking Nhe1 and BamH1 restriction sites |
| Transfected construct (*H. sapiens*) | gE-CD4 | gBlock by IDT | US8 of MN136524.1 fused to human CD4 TM and cytosolic tail BT019791.1 | Cloning via added flanking Nhe1 and BamH1 restriction sites |
| Antibody | Cytotect Human plasma pool polyclonal | Biotest | | Titration as indicated in this study, 1:100 in flow cytometry |
| Antibody | αCD107a-APC mouse monoclonal | BD FastImmune | Clone H4A3 | 1:50 in flow cytometry |
| Antibody | αCD56-BV650 mouse monoclonal | Biolegend | Clone 5.1H11 | 1:50 in flow cytometry |
| Antibody | αCD3-FITC mouse monoclonal | Biolegend | Clone UCHT1 | 1:50 in flow cytometry |
| Antibody | αhuman-IgG-PE mouse monoclonal | BD | | 1:100 in flow cytometry |
| Antibody | αHis-PE mouse monoclonal | Miltenyi Biotec | | 1:100 in flow cytometry |
| Peptide, recombinant protein | Human Fcγ-TexasRed | Rockland | | Human IgG-Fc fragment |
| Antibody | Rituximab Humanized monoclonal | Roche, University Hospital Freiburg Pharmacy | | Titration as indicated in this study, 1:100 in flow cytometry |

*Continued on next page*

*Continued*

| Reagent type (species) or resource | Designation | Source or reference | Identifiers | Additional information |
|---|---|---|---|---|
| Antibody | Herceptin Humanized monoclonal | Roche, University Hospital Freiburg Pharmacy | | Titration as indicated in this study |
| Antibody | αCD20-PE mouse monoclonal | Miltenyi Biotec | | 1:100 in flow cytometry |
| Antibody | αhuman IgG-FITC Polyclonal rabbit | ThermoFisher | | 1:100 in flow cytometry |
| Antibody | THE Anti-His-HRP mouse monoclonal | Genscript | | 0.5 µg/ml in ELISA |
| Antibody | MSL-109 Humanized monoclonal | Absolute antibody | | 10 µg/ml in flow cytometry |
| Antibody | B12 Humanized monoclonal | Kindly provided by Ann Hessell, Scripps USA | | 1 µg/ml in precipitation |
| Antibody | B12 LALA Humanized monoclonal | Kindly provided by Ann Hessell, Scripps USA | | 1 µg/ml in precipitation |
| Recombinant DNA reagent | pIRES-eGFP | Addgene | | |
| Recombinant DNA reagent | pSLFRTKn | doi:10.1128/jvi.76.17.8596-8608 | | |
| Sequence-based reagent | KL-Delta TRL11-Kana1 | This paper | PCR primer | ACGACGAAGAGG ACGAGGACGACAA CGTCTGATAAGGAA GGCGAGAACGTGT TTTGCACCCCAGTG AATTCGAGCTC GGTAC |
| Sequence-based reagent | KL-DeltaTRL11-Kana2 | This paper | PCR primer | TGTATACGCCGT ATGCCTGTACGTGA GATGGTGAGGTCTT CGGCAGGCGACACG CATCTTGACCATGA TTACGCCAAGCTCC |
| Sequence-based reagent | KL-DeltaTRL12-Kana1 | This paper | PCR primer | CGGACGGACCTAG ATACGGAACCTTTG TTGTTGACGGTGGA CGGGGATTTACAG TAAAAGCCAGTGAA TTCGAGCTCGGTAC |
| Sequence-based reagent | KL-DeltaTRL12-Kana2 | This paper | PCR primer | CCTTACAGAATGT TTTAGTTTATTGTT CAGCTTCATAAGAT GTCTGCCCGGAAA CGTAGCGACCATGA TTACGCCAAGCTCC |
| Sequence-based reagent | KL-DeltaUL119-Kana1 | This paper | PCR primer | TTGTTTATTTTGT TGGCAGGTTGGC GGGGGAGGAAAA GGGGTTGAACAG AAAGGTAGGTGC CAGTGAATTCG AGCTCGGTAC |

*Continued on next page*

*Continued*

| Reagent type (species) or resource | Designation | Source or reference | Identifiers | Additional information |
|---|---|---|---|---|
| Sequence-based reagent | KL-DeltaUL119-Kana2 | This paper | PCR primer | AGGTGACGCGAC CTCCTGCCACATA TAGCTCGTCCAC ACGCCGTCTCGTC ACACGGCAACGA CCATGATTACG CCAAGCTCC |
| Peptide, recombinant protein | sgp68-V5/His | This paper | Ectodomain of UL119-118 from MN900952.1 | V5/His6-tagged. Produced in 293T cells |
| Peptide, recombinant protein | sgp34-V5/His | This paper | Ectodomain of RL11 from MN900952.1 | V5/His6-tagged. Produced in 293T cells |
| Peptide, recombinant protein | sgp34mtrp-V5/His | This paper | Ectodomain of RL11 mtrp mutant from MN900952.1 | V5/His6-tagged. Produced in 293T cells |
| Peptide, recombinant protein | sgp68-strep | This paper | Ectodomain of UL119-118 from MN900952.1 | Streptavidin-tagged. Produced in 293T cells |
| Peptide, recombinant protein | sgp34-strep | This paper | Ectodomain of RL11 from MN900952.1 | Streptavidin-tagged. Produced in 293T cells |
| Peptide, recombinant protein | sgp34mtrp-strep | This paper | Ectodomain of RL11 mtrp mutant from MN900952.1 | Streptavidin-tagged. Produced in 293T cells |
| Peptide, recombinant protein | CD16-Avi/His | Sino Biological | Soluble recombinant FcγRIIIA | Avi/His-tagged 5 µg/ml in flow cytometry |
| Peptide, recombinant protein | CD32A-Avi/His | Sino Biological | Soluble recombinant FcγRIIA | Avi/His-tagged 5 µg/ml in flow cytometry |
| Peptide, recombinant protein | CD32B-Avi/His | Sino Biological | Soluble recombinant FcγRIIB | Avi/His-tagged 5 µg/ml in flow cytometry |
| Peptide, recombinant protein | CD64-Avi/His | Sino Biological | Soluble recombinant FcγRI | Avi/His-tagged 5 µg/ml in flow cytometry |
| Peptide, recombinant protein | wtFc | Kindly provided by Pamela Bjorkman, Caltech USA doi:10.1128/JVI.01476-07 | | |
| Peptide, recombinant protein | nbFc | Kindly provided by Pamela Bjorkman, Caltech USA doi:10.1128/JVI.01476-07 | | |
| Commercial assay or kit | Pierce F(ab')2 Preparation Kit | ThermoFisher | | |
| Commercial assay or kit | Easytag Express | Perkin Elmer | | |
| Software, algorithm | Prism | GraphPad | | |
| Software, algorithm | FlowJo | BD | | |

## Cell culture

All cells were cultured in a 5% $CO_2$ atmosphere at 37℃. MRC-5 (ECACC 05090501), HFF (HF-99/7 kindly provided by Dieter Neumann-Haefelin and Valeria Kapper-Falcone, Institute of Virology, Freiburg Germany), 293 T-CD20 (kindly provided by Irvin Chen, UCLA USA [*Morizono et al., 2010*]), BJ-Her2 (BJ-5ta foreskin fibroblasts [ATCC CRL-4001] stably expressing Her2/Erbb2 (NM_004448), lentiviral transduction as in *Halenius et al., 2011*), and Hela cells (ATCC CCL-2) were maintained in Dulbecco's modified Eagle's medium (DMEM, Gibco) supplemented with 10% (vol/vol) fetal calf serum (FCS, Biochrom). BW5147 mouse thymoma cells (kindly provided by Ofer Mandelboim, Hadassah Hospital, Jerusalem, Israel) were maintained at $3 \times 10^5$ to $9 \times 10^5$ cells/ml in Roswell Park Memorial Institute medium (RPMI GlutaMAX, Gibco) supplemented with 10% (vol/vol) FCS, sodium pyruvate (1×, Gibco) and β-mercaptoethanol (0.1 mM, Sigma). Cells were used when tested negative for mycoplasma contamination (Eurofins). If positive, cells were treated using Plasmocure (Invivogen) or BM-Cyclin (Roche) as instructed by the supplier.

## Molecular cloning and genetic constructs

Sequences for HCMV *RL11* (gp34) and HCMV *UL119-118* (gp68) were taken from according sequences in p_BAC2 AD169 (MN900952.1) and synthesized as gBlocks (IDT) flanked with *Nhe1* and *BamH1* restriction sites suitable for insertion into pIRES_eGFP (Addgene). Sequences for HSV-1 *US8* (gE) and HSV-1 *US7* (gI) were taken from HSV-1 (strain F), synthesized as gBlocks and cloned as above. Human CD4 transmembrane and cytosolic domain (GenBank: M35160) was used to substitute the transmembrane and cytosolic domain of respective vFcγR sequences. vFcγR-CD4 fusion constructs were synthesized as gBlocks and cloned as above. Her2 antigen was acquired from Addgene (pCDNA3). CD99 (GenBank: BC010109) was cloned into pIRES_eGFP via Nhe1 and BamH1 restriction sites. Flanking restriction sites were introduced via PCR (Primers: IDT). GOI-internal BamH1 sites were removed by silent single nucleotide substitutions during gBlock design.

## Recombinant vaccinia virus (rVACV)

The construction of the rVACVs has been described before (*Atalay et al., 2002*; *Sprague et al., 2008*; *Staib et al., 2004*). In brief, the gene of interest inserted in the vaccinia virus recombination vector p7.5k131a was transferred into the thymidine kinase open reading frame of the virus genome (strain Copenhagen). rVACVs were selected with bromodeoxyuridine (BrdU; 100 μg/ml) using tk-143 cells (ATCC CVCL_2270).

## Metabolic labeling and human Fcγ precipitation

Metabolic labeling using Easytag Express [$^{35}$S]-Met/Cys protein labeling, Perkin Elmer with 100 Ci/ml for 2 hr, and immunoprecipitation of Fcγ fragments using CNBr-Sepharose (GE Healthcare) was performed as described previously (*Sprague et al., 2008*). Generation and purification of human Fcγ fragments wtFc and nbFc are described elsewhere (*Sprague et al., 2004*). In brief, wild-type and mutant IgG Fc proteins were collected from CHO cell supernatants and purified using $Ni^{2+}$-NTA affinity chromatography followed by pH sensitive FcRn-Sepharose column separation and subsequent size exclusion chromatography. B12 and B12-LALA were kind gifts from Ann Hessell (*Hessell et al., 2007*). Direct IgG precipitation was performed using Protein G Sepharose (Amersham). Samples were de-glycosylated using EndoH (NEB, as suggested by the supplier).

## Production of soluble HCMV vFcγRs

Expression constructs encoding tagged HCMV vFcγR constructs lacking their respective transmembrane and cytosolic domains were generated as described elsewhere (*Corrales-Aguilar et al., 2014b*). Soluble C-terminally V5-His-tagged vFcγR molecules were produced by transfection of 293 T cells in a 10 cm dish format and covered with 7 ml of DMEM/5% FCS. 3 days post transfection the supernatants were harvested, centrifuged (11,000 g, 30 min), and used directly or stored in 1 ml aliquots at −20℃. Production of soluble vFcγRs was controlled via α-tag immunoblot.

## Generation of HCMV BAC2 derived virus mutants

Recombinant HCMV mutants were generated according to previously published procedures (*Tischer et al., 2006*; *Wagner et al., 2002*) using pAD169-BAC2 (MN900952.1, *Le-Trilling et al.,*

*2020*) corresponding to AD169varL (*Le et al., 2011*) as parental genome. For the construction of the HCMV deletion mutants, a PCR fragment was generated using the plasmid pSLFRTKn (*Atalay et al., 2002*) as the template DNA. The PCR fragment containing a kanamycin resistance gene was inserted into the parental BAC by homologous recombination in *E. coli*. The inserted cassette replaces the target sequence which was defined by flanking sequences in the primers. This cassette is flanked by *frt*-sites which can be used to remove the kanamycin resistance gene by *FLP*-mediated recombination. The removal of the cassette results in a single remaining *frt*-site. The deletion of multiple non-adjacent genes was conducted in consecutive steps.

The gene *TRL11* was deleted by use of the primers KL-DeltaTRL11-Kana1 (ACGACGAAGAG-GACGAGGACGACAACGTCTGATAAGGAAGGCGAGAACGTGTTTTGCACCCCAGTGAATTCGAGC TCGGTAC) and KL-DeltaTRL11-Kana2 (TGTATACGCCGTATGCCTGTACGTGAGATGGTGAGGTC TTCGGCAGGCGACACGCATCTTGACCATGATTACGCCAAGCTCC). The gene TRL12 was deleted by use of the primers KL-DeltaTRL12-Kana1 (CGGACGGACCTAGATACGGAACCTTTGTTG TTGACGGTGGACGGGGATTTACAGTAAAAGCCAGTGAATTCGAGCTCGGTAC) and KL-Del-taTRL12-Kana2 (CCTTACAGAATGTTTTAGTTTATTGTTCAGCTTCATAAGATGTCTGCCCGGAAACG TAGCGACCATGATTACGCCAAGCTCC). The gene UL119 was deleted by use of the primers KL-DeltaUL119-Kana1 (TTGTTTATTTTGTTGGCAGGTTGGCGGGGGAGGAAAAGGGGTTGAACA-GAAAGGTAGGTGCCAGTGAATTCGAGCTCGGTAC) and KL-DeltaUL119-Kana2 (AGGTGACGC-GACCTCCTGCCACATATAGCTCGTCCACACGCCGTCTCGTCACACGGCAACGACCATGA TTACGCCAAGCTCC).

## Ni²⁺-NTA Co-precipitation

10 ml soluble vFcγR supernatants were mixed 1:1 in the presence of 1 µg Rituximab. Antibody has to be given in a limiting amount to increase the probability of simultaneous binding to the same IgG molecule. Samples were incubated 1 hr at 4°C and then mixed with freshly prepared $Ni^{2+}$-NTA Sepharose beads (2 µl BV, cOmplete His-Tag purification resin) and incubated overnight at 4°C in the presence of 20 mM Imidazole (rotate or shake). Beads were washed three times with PBS/20 mM Imidazole at 11,000 g and 4°C. After the final wash, beads were either resuspended in sample buffer (Tris pH 6.8, SDS, Glycerol, 2-mercaproethanol, bromophenol blue) or subjected to a PNGase F digest before being supplemented with sample buffer (NEB, performed as suggested by the supplier). Samples were then denatured at 95°C and analyzed via SDS-PAGE and subsequent immunoblot.

## vFcγR-binding ELISA

We adapted a standard ELISA protocol to measure binding between soluble vFcγRs and target IgG. 96-well Nunc Maxisorp plates were coated with 1 µg of biotin in PBS. Plates were then blocked and incubated with titrated amounts of supernatants from soluble strep-tagged vFcγR producing cells (1° vFcγRs). Supernatants were diluted in PBS. After incubation with 100 ng/well of target antibody, plates were incubated with a 2° His-tagged soluble vFcγR (1:2 diluted) followed by detection using a HRP-conjugated anti-His antibody. Plates were measured using a Tecan Genios Pro microtiter plate reader at 450 nm/630 nm. Plates were washed three times between steps using PBS/0.05% Tw-20.

## FcγR-binding assay

Transfected or infected vFcγR expressing cells were harvested using Accutase (Sigma-Aldrich) to retain surface molecules upon detachment. Harvested cells were washed in PBS, equilibrated in staining buffer (PBS, 3% FCS) and sedimented at 1000 g and 10°C for 3 min. Cells were then incubated with staining buffer containing rituximab, Cytotect, or herceptin. Cells were then incubated in an adequate volume of staining buffer containing either mAbs, Fcγ fragment, or FcγR ectodomains pre-incubated with an αHis-PE antibody (30 min, 4°C). Human His-tagged FcγR ectodomains were used at a final concentration of 5 µg/ml (1:50 dilution from reconstituted 0.25 mg/ml stock solution; Sino Biological, 10389-H08H1). Pre-incubation with Protein G (Rockland, Biotin conjugated) was performed prior to incubation with FcγR ectodomains (diluted 1:100). Further incubation steps were carried out at 4°C for 1 hr and followed by three washing steps in staining buffer. Dead cells were excluded via DAPI stain. Analysis was performed on a FACS Fortessa instrument (BD Bioscience).

### IC internalization assay

Transfected 293 T-CD20 cells were harvested using Accutase (Sigma-Aldrich) to retain surface molecules upon detachment. Harvested cells were incubated with Rituximab (1 μg/well) for 1 hr at 4°C in medium. Cells were then washed twice in medium containing 5% FCS and seeded into a 96-well plate at $2 \times 10^4$ cells/well. Each reaction was performed on cells from one 96-well. Cells were then incubated at 37°C in a 5% $CO_2$ atmosphere until being harvested at different time points ensuring regular re-suspension to avoid cell attachment over longer periods of time. After harvesting, cells were directly stained with αhuman-IgG-PE for 1 hr at 4°C and fixed using 3% PFA. Analysis was performed on a FACS Fortessa instrument (BD Bioscience).

### FcγR activation assay

The assay was performed as described earlier (*Corrales-Aguilar et al., 2013*). Briefly, target cells were incubated with titrated amounts of antibody (96-well format) in medium (DMEM) supplemented with 10% FCS for 30 min at 37°C, 5% $CO_2$. Cells were washed with medium (RPMI) and co-cultured with BW5147-reporter cells (ratio E:T 20:1) expressing individual host FcγR ectodomains or CD99 as control for 16 hr at 37°C in a 5% $CO_2$ atmosphere. Reporter cell mIL-2 secretion was quantified by subsequent anti IL-2 sandwich ELISA as described previously (*Corrales-Aguilar et al., 2013*).

### NK cell degranulation assay

PBMCs were purified from healthy donor blood by centrifugation via Lymphoprep Medium according to the supplier instructions (Anprotec). HCMV-infected MRC-5 or HFF cells (MOI = 3, 72hpi) were incubated with titrated amounts of Cytotect at 37°C for 1 hr in a 5% $CO_2$ atmosphere. After washing $2\times$ with medium, $5 \times 10^5$ PBMCs were incubated on opsonized infected cells for 6 hr (100 μl per well in RPMI/10% FCS) in the presence of αCD107a-, αCD56-BV650, and Golgi-Plug/Golgi-Stop (according to supplier, BD). After incubation, PBMCs were harvested, washed in staining buffer (PBS/3% FCS), and incubated with αCD3-FITC 30 min at 4°C. After two final washing steps in staining buffer, analysis was performed on a FACS Fortessa instrument (BD Bioscience).

### Commercial antibody preparations

Antibodies were diluted 1:100 on $1 \times 10^6$ cells for flow cytometry; Cytotect (Biotest); αCD107a-APC (BD FastImmune clone H4A3); αCD56-BV650 (Biolegend clone 5.1H11); αCD3-FITC (Biolegend clone UCHT1); αhuman-IgG-PE (BD); human Fcγ-TexasRed (Rockland); rituximab (Rtx, Roche); herceptin (Hc, Roche); αhuman-IgG-PE (Miltenyi Biotec); αHis-PE (Miltenyi Biotec); αCD20-PE (Miltenyi Biotec); polyclonal rabbit-αhuman IgG-FITC (ThermoFisher); THE Anti-His-HRP (Genscript); MSL-109 anti HCMV gH (Absolute Antibody).

### Statistical analyses

Statistical analyses were performed using ANOVA or t-test (Prism5, Graphpad). Multiple comparison was corrected by Tukey test.

## Acknowledgements

We are grateful to Ann Hessell (The Scripps Research Institute, La Jolla, California, USA) for providing antibody B12 and B12-LALA and Irvin Chen, UCLA for providing 293 T-CD20 cells. We thank Pamela Bjorkman (California Institute of Technology, Pasadena, California, USA) for providing the wtFc and nbFc fragments.

## Additional information

### Funding

| Funder | Grant reference number | Author |
| --- | --- | --- |
| Deutsche Forschungsge-meinschaft | HE2526/9-1 | Hartmut Hengel |
| Bundesministerium für Bildung | 031L0090 | Hartmut Hengel |

und Forschung

| Albert-Ludwigs-Universität Freiburg | EQUIP - Funding for Medical Scientists | Philipp Kolb |
|---|---|---|
| Deutsche Forschungsgemeinschaft | HA6035/2-1 | Anne Halenius |
| Deutsche Forschungsgemeinschaft | FOR2830 | Hartmut Hengel |
| Deutsche Forschungsgemeinschaft | FOR2830 | Anne Halenius |

The funders had no role in study design, data collection and interpretation, or the decision to submit the work for publication.

### Author contributions

Philipp Kolb, Conceptualization, Data curation, Formal analysis, Investigation, Visualization, Writing - original draft; Katja Hoffmann, Data curation, Formal analysis, Investigation, Writing - review and editing; Annika Sievert, Henrike Reinhard, Eva Merce-Maldonado, Data curation, Investigation; Vu Thuy Khanh Le-Trilling, Resources, Methodology, Writing - review and editing; Anne Halenius, Resources; Dominique Gütle, Data curation, Formal analysis; Hartmut Hengel, Conceptualization, Supervision, Funding acquisition, Writing - original draft

### Author ORCIDs

Philipp Kolb https://orcid.org/0000-0001-7935-217X
Hartmut Hengel https://orcid.org/0000-0002-3482-816X

### Ethics

Human subjects: Consent of blood donors was approved by the ethical review committee, University of Freiburg, vote 474/18.

### Decision letter and Author response

Decision letter https://doi.org/10.7554/eLife.63877.sa1
Author response https://doi.org/10.7554/eLife.63877.sa2

# Additional files

### Supplementary files

• Transparent reporting form

### Data availability

All data generated or analysed during this study are included in the manuscript and supporting files. Source data files have been provided for all pertinent Figures (Figures 1D, 3D, 4C, 5D, 6B, 6C, 6D, Figure 4—figure supplement 1).

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
