## [Decision Letter]

**Acceptance summary:**

This work gives important novel insights into immune evasion strategies by Human Cytomegalovirus with a focus on evasion from antibody mediated immune control. The authors show that HCMV counteracts Fc-gamma receptor (FcγR) mediated immune control through the viral Fc-gamma binding glycoproteins (vFcγRs) gp34 and gp68. Those viral proteins bind IgG simultaneously at topologically different Fcγ sites, and thereby efficiently antagonize FcγR activation of the host.

**Decision letter after peer review:**

Thank you for submitting your article "Human Cytomegalovirus antagonizes activation of Fcγ receptors by distinct and synergizing modes of IgG manipulation" for consideration by *eLife*. Your article has been reviewed by three peer reviewers, and the evaluation has been overseen by a Reviewing Editor and Päivi Ojala as the Senior Editor. The reviewers have opted to remain anonymous.

The reviewers have discussed the reviews with one another and the Reviewing Editor has drafted this decision to help you prepare a revised submission.

Summary:

All three reviewers agreed that this work is very solid and interesting, giving important novel insights into immune evasion strategies by Human Cytomegalovirus with a focus on evasion from antibody mediated immune control. The authors show that HCMV counteracts Fc-gamma receptor (FcγR) mediated immune control through the viral Fc-gamma binding glycoproteins (vFcγRs) gp34 and gp68. Those viral proteins bind IgG simultaneously at topologically different Fcγ sites, and thereby efficiently antagonize FcγR activation of the host.

During our internal discussion, all three reviewers also agreed that the manuscript text shows some clear limitations. The text either needs to be adjusted or, if the authors insist on the cooperativity between gp34 and gp68, additional experiments need to be performed as outlined by reviewers 1 and 2. We all agreed to leave this decision to the authors.

Essential Revisions:

1) The data presented in gp34 and gp68 transfected cells demonstrate some cooperativity with FcR reactivation. However, the addition of a high concentration of pooled anti-CMV IgGs to virus infected cells using virus variants lacking gp34, gp68, or both gp34/gp68 did not show cooperativity. This point is very important for the manuscript, as it is currently written, and therefore requires modification of the text, adjustment of the conclusions, or additional experiments.

2) Many figures only show, at best, a 2 fold change and it should be discussed that limiting ADCC by these viral genes is not a major immune evasion strategy.

3) The nature of the viruses needs to be described in more detail/more clearly. The text, in its current form, overlooks the absence of all other viral FcRs as they are assessing the cooperativity of gp34/gp68.

Reviewer #1:

Kolb et al., address a long-standing question regarding the human cytomegalovirus (HCMV) IgG-Fc-binding functions. Two known virus-encoded Fc-gamma-binding receptors (vFcgRs), vRL11-encoded gp34 and UL119-118-encoded gp68 were previously shown by this group to prevent activation of cellular receptors for IgG. This manuscript now shows that these two synergize in antagonizing host FcgR activation such that "gp34 enhances immune complex internalization and gp68 acts as inhibitor of host FcγR binding to immune complexes", revealing that these viral mimic block conformational changes within IgG that are crucial for Fc-mediated effector function. The major limitation of the work is the use of a parental virus that lacks other vFcgRs, which likely provides unencumbered phenotypes for gp34 and gp68 but is not the case with wild strains of HCMV. The limitations of the assay viruses must be fully disclosed and addressed in the manuscript.

Figure 1 clearly shows that the two viral proteins bind independently to IgG, with gp34 associating with the upper hinge region and gp68 associating with the immunoglobulin domain CH2-CH3 interface. The data are consistent with previous literature and are themselves clear and convincing.

Figure 2 seeks to demonstrate that antagonism by vFcgRs is cooperative using HCMV mutant viruses that are deficient in one or both, experiments performed a background of ∆RL12 (gp95-deficiency). AD169 lacks many genes that are naturally found in wild strains of HCMV and that might influence the results observed here, including RL13 (gpRL13) which is lost in every viral strain upon propagation in cultured cells. Thus, panel A labeling requires much more text support owing to the use of not just ∆RL12 but ∆RL12/∆RL13 (and whatever other genes are absent in AD169 background). For clarity, ∆RL11 (gp34-deficient), ∆UL119-118 (gp68-deficient) and ∆RL11/∆RL119-118 (double-deficient) terminology might be adopted for clarity. (NOTE: It would be important here to mention any specific role of UL12 (gp95) because mutant viruses appear to be RL11/RL12 double mutants. The same goes for the fact that RL13 is absent in this virus strain, although that might be an intractable issue.) Ambiguities would be resolved with experiments in a more fully competent strain of HCMV (such as in the conditional Merlin strain that retains RL13 as well as other genes that have gone missing in AD169). Further, statistical evaluations are necessary here to be sure that the appearance of a reduction in IL-2 production occurs only with the double mutant, and the y-axis should be labeled with the biologic protein being measured by ELISA as a surrogate for FcgRIII activation. Finally, the text should explain to the reader why antagonism by parental and either "single mutant" is associated with higher levels of IL-2 and lower NK cell CD107a positivity (panel B), which seem counterintuitive to this reviewer. These data seem to compare relevant mutants but do not allow the conclusion that "that gp34 and gp68 antagonize FcγRs by cooperative modes of action" without stating as well "in the absence of other known vFcgRs encoded by RL12 and RL13. Finally, careful work with viral mutants in the absence of biophysical data makes any conclusion of "cooperative modes of action" premature.

Figure 3 examines the dynamics of gp34 and gp68 internalization, after replacing the transmembrane and cytosolic domains with CD4 domains, where differences compatible with the interpretation, "gp34 compared to gp68 is mediating the efficient internalization of IC" might be expressed better as "native gp34 appears to be more efficient at internalizing IC than native gp68". Here, it is not clear why more work was not performed with a mix of native gp34 and CD4 modified gp68 as well as native gp68 and CD4 modified gp34 (with the mix of and CD4 modified gp68 and CD4 modified gp34 potentially useful as well). There may be cooperation here and the data presented only hint at the conclusion that the authors favor.

Figure 4 makes further use of CD4 modified gp34 and gp68 to maintain cell surface expression, and the conclusion, "gp68 but not gp34 to significantly reduce binding of FcγRIIIA/B to cell surface immune complexes compared to CD99" is warranted. Again, it seems combination of native proteins with CD4 modified proteins would add important information to the interpretations offered here.

Figure 5 returns to virus-infected cells to evaluate "a strong relative reduction in FcγRIIIA/B binding to opsonized HCMV-infected cells in the presence of gp68, but not gp34", which is substantiated by the data shown.

Figure 6 addresses the issue of why authors "did not observe efficient antagonization of FcγR activation by gp34 or gp68 individually in the context of HCMV deletion mutant infection". Using an assay set up in HeLa cells, "gp68 or gp34 individually confirmed to antagonize FcγRIII activation", "the more membrane resident gp68-CD4 showed a markedly stronger antagonistic effect compared to gp68wt", and "internalization to be a major condition of gp34 driven antagonization of FcγR activation", which are substantiated. The authors then observe that "the presence of an excess of non-immune IgG interferes with both gp68 and gp34 antagonization" that is "attributed to displacement of immune IgG from the vFcγRs resulting in restoration of FcγRIII activation". All of this data appears useful and appropriate.

Figure 7 goes on to show that the "ectodomain of gp68 is sufficient to enhance the antagonistic effect of gp34", which gets at some of the concerns raised with earlier data where combinations would have probably brought this to light.

The overall conclusions are warranted, so long as it is clear to the reader that the virus being used lacks all other potentially relevant vFCgRs (when the double mutant is introduced, it would be appropriate to state that it lacks all known vFcgRs). In addition, it would be appropriate to divulge whether elimination of UL12 is necessary for these observations even if studies with conditional RL13 expression are not readily approachable and remains a mystery for future consideration. Limitations of the parental virus backbone should be an important topic for the Discussion. It is unlikely in this reviewer's eye that a set of mutants in fully WT (conditional) virus would behave the same as these particular set are shown to behave, thus discussion relating to clinical issues should be toned down.

Reviewer #2:

The manuscript investigates the mechanism cytomegalovirus proteins gp68 and gp34 as an immune evasion strategy to limit Fc receptor-mediated antibody-dependent cell-mediated cytotoxicity. The study is based on previous findings characterizing gp34 and gp68 as viral evasins during a virus infection. The manuscript describes that gp34 and gp68 function in a synergistic manner by enhances immune complex internalization and as an inhibitor of FcγR binding to immune complexes, respectively. The manuscript systematically examines the function of gP34 and gp68 through virus infection and protein expression in transfected cells. Yet, the manuscript conclusion is based on only two-fold differences with the key conditions and virus studies do not demonstrate synergistic findings (Figure 2).

1) Figure 2 is a key experiment that demonstrates that the lack of gp34 and gp68 have an impact on FcyR activation. The data does not show cooperativity to down limit FcyR activation. The study should expand the kinetics of study and different concentrations of Cytotect as well as specific anti-CMV antibodies.

2) The study is based on previous publications and further define the mechanism of actions. However, the findings typically only demonstrate at most a 2-fold difference upon the expression of gp34 and gp68. This is mostly due to the analysis of using flow cytometry. The authors should include a more sensitive assay to define the function of gp34 and gp68 on FcyR activation.

Reviewer #3:

The manuscript by Kolb et al. builds on this laboratories 2014 publication showing that HCMV gp34 and gp68 are able to bind Fcg and in doing so, interrupt FcgR mediated activation signals including ADCC by NK cells. The paper demonstrates that gp34 and gp68 bind IgG at the same time and in doing so act in concert to mediate efficient prevention of FcgR activation by preventing cellular FcgR from binding the immune IgG and the removal of immune IgG/HCMV protein immune complexes from the cell surface, mediated by gp34. The apparent redundancy seen with HCMV immune evasion mechanisms is interesting and the clear indication from this work is that what might appear redundancy is actually not and more to do with efficient biological function with 2 viral proteins working in conjunction.

The manuscript is on the whole well written and the experiments set out in a logical fashion. The inclusion of cartoons to assist the reader in understanding the experimental set up is welcome. The results demonstrate simultaneous binding that the dual action of gp34/68 is required for efficient inhibition of FcgR activation the role of gp34 in internalization and then the role gp68 in FcgR blocking. While various antigen v FcgR models are used for these experiments HCMV infected cells have also been used. The effect of excess non HCMV immune IgG as would be seen in sera/ in vivo has also been considered.

The experiments have been carefully thought through and are well controlled appropriate statistical analysis has been performed and the data presented support the authors conclusions. I do not have any major concerns over this work.

---

## [Author Response]

Essential Revisions:1) The data presented in gp34 and gp68 transfected cells demonstrate some cooperativity with FcR reactivation. However, the addition of a high concentration of pooled anti-CMV IgGs to virus infected cells using virus variants lacking gp34, gp68, or both gp34/gp68 did not show cooperativity. This point is very important for the manuscript, as it is currently written, and therefore requires modification of the text, adjustment of the conclusions, or additional experiments.

We thank the reviewers for addressing this important point regarding our experiments in the context of viral infection when using HCMV hyperimmunoglobulin which contains excess of non-immune IgG (see Figure 2). As we here observed a strong antagonization when both molecules are expressed, but not when either gp34 or gp68 are expressed alone, we hypothesized that there must be cooperative mode of action. We agree with reviewer that the observed synergism does not allow the conclusion that there is a mechanistic cooperation from different effects. To address the concerns of the reviewers, we changed the text and figure legend to not overstate the data and describe the effects seen in Figure 2 as “synergizing”. This adjustment expresses that we do not explicitly show cooperation in the sense of a molecular mechanism, but that a more than additive effect is observed. As the data strongly suggests cooperation, we then address this in the following experiments. In settings of reduced complexity, we finally confirmed cooperation and identified the opposing role of non-immune IgG as outlined in Figure 6. We added a new experiment in Figure 6C, showing synergy also in absence of non-immune IgG using a the mAb Herceptin in the context of HCMV infection by infecting newly generated Her2-expressing fibroblasts (see Results). This experiment also further supports the conclusions made in Figure 6B, that gp34 and gp68 individually show antagonization. Due to this experiment, Anne Halenius was added as a contributing author.

2) Many figures only show, at best, a 2 fold change and it should be discussed that limiting ADCC by these viral genes is not a major immune evasion strategy.

We agree that some experiments show weak effects when looking at single values along a titration curve. We now express the overall antagonistic effects measured by our FcR-reporter assay as “area under curve” values with added statistics (see Figure 2). This expresses the antagonistic effect of vFcRs over the whole titration curve compared to single IVIg dilutions. This type of evaluation considers differences in the presence of excess non-immune IgG and better illustrates the strength of the antagonistic effects mediated by gp34 and gp68.

3) The nature of the viruses needs to be described in more detail/more clearly. The text, in its current form, overlooks the absence of all other viral FcRs as they are assessing the cooperativity of gp34/gp68.

We agree that the decision to exclude other vFcRs has to be explained in more detail. In addition, we now clearly state the nature of the viral deletion mutants used in Figure 2 and the referring text and justify our decision. In brief, as we were specifically interested in the mechanisms behind the FcR-antagonists gp34 and gp68, we chose to exclude gp95 and gpRL13. Major arguments behind this decision were i) that gpRL13 is notoriously mutated in the HCMV genome upon cell culture passages and has not yet been described to antagonize FcR activation and ii) that gp95 is the most polymorphic gene of HCMV with only minor surface expression in the context of AD169 infection (Corrales-Aguilar et al., 2014). Due to remarkably little conservation of gp95 allelic forms across the strains of HCMV and its variable efficacy to inhibit host FcγRs, we did not consider this molecule when describing the mechanisms of the much more conserved and well-described vFcRs gp34 and gp68. A manuscript comparing gp95 alleles is currently in preparation.

Reviewer #1:Kolb et al., address a long-standing question regarding the human cytomegalovirus (HCMV) IgG-Fc-binding functions. Two known virus-encoded Fc-gamma-binding receptors (vFcgRs), vRL11-encoded gp34 and UL119-118-encoded gp68 were previously shown by this group to prevent activation of cellular receptors for IgG. This manuscript now shows that these two synergize in antagonizing host FcgR activation such that "gp34 enhances immune complex internalization and gp68 acts as inhibitor of host FcγR binding to immune complexes", revealing that these viral mimic block conformational changes within IgG that are crucial for Fc-mediated effector function. The major limitation of the work is the use of a parental virus that lacks other vFcgRs, which likely provides unencumbered phenotypes for gp34 and gp68 but is not the case with wild strains of HCMV. The limitations of the assay viruses must be fully disclosed and addressed in the manuscript.Figure 1 clearly shows that the two viral proteins bind independently to IgG, with gp34 associating with the upper hinge region and gp68 associating with the immunoglobulin domain CH2-CH3 interface. The data are consistent with previous literature and are themselves clear and convincing.Figure 2 seeks to demonstrate that antagonism by vFcgRs is cooperative using HCMV mutant viruses that are deficient in one or both, experiments performed a background of ∆RL12 (gp95-deficiency). AD169 lacks many genes that are naturally found in wild strains of HCMV and that might influence the results observed here, including RL13 (gpRL13) which is lost in every viral strain upon propagation in cultured cells. Thus, panel A labeling requires much more text support owing to the use of not just ∆RL12 but ∆RL12/∆RL13 (and whatever other genes are absent in AD169 background). For clarity, ∆RL11 (gp34-deficient), ∆UL119-118 (gp68-deficient) and ∆RL11/∆RL119-118 (double-deficient) terminology might be adopted for clarity. (NOTE: It would be important here to mention any specific role of UL12 (gp95) because mutant viruses appear to be RL11/RL12 double mutants. The same goes for the fact that RL13 is absent in this virus strain, although that might be an intractable issue.) Ambiguities would be resolved with experiments in a more fully competent strain of HCMV (such as in the conditional Merlin strain that retains RL13 as well as other genes that have gone missing in AD169). Further, statistical evaluations are necessary here to be sure that the appearance of a reduction in IL-2 production occurs only with the double mutant, and the y-axis should be labeled with the biologic protein being measured by ELISA as a surrogate for FcgRIII activation. Finally, the text should explain to the reader why antagonism by parental and either "single mutant" is associated with higher levels of IL-2 and lower NK cell CD107a positivity (panel B), which seem counterintuitive to this reviewer. These data seem to compare relevant mutants but do not allow the conclusion that "that gp34 and gp68 antagonize FcγRs by cooperative modes of action" without stating as well "in the absence of other known vFcgRs encoded by RL12 and RL13. Finally, careful work with viral mutants in the absence of biophysical data makes any conclusion of "cooperative modes of action" premature.

We thank the reviewer for his/her thoughtful and very thorough evaluation of our data.

– Figure 2A: labeling was made more clear and the genetic background of the virus mutants used was clarified in the text and the legend. The designation of the mutants was kept throughout all figures (Figure 5A and Figure 6C).

– Figure 2A: y-Axis labeling now includes “IL-2 A450nm”.

– Figure 2A: “Area under curve” evaluation was added and statistical analysis was performed.

– The conclusions were changed to “synergize” instead of “cooperate” to not overstate the data at this point. We make this conclusion subject to the absence of gp95 (RL12) and gpRL13 (RL13) (see Results). Cooperation between gp34 and gp68 and the role of non-immune IgG is specifically addressed in Figure 6.

Figure 3 examines the dynamics of gp34 and gp68 internalization, after replacing the transmembrane and cytosolic domains with CD4 domains, where differences compatible with the interpretation, "gp34 compared to gp68 is mediating the efficient internalization of IC" might be expressed better as "native gp34 appears to be more efficient at internalizing IC than native gp68". Here, it is not clear why more work was not performed with a mix of native gp34 and CD4 modified gp68 as well as native gp68 and CD4 modified gp34 (with the mix of and CD4 modified gp68 and CD4 modified gp34 potentially useful as well). There may be cooperation here and the data presented only hint at the conclusion that the authors favor.

– Wording was optimized to distinguish the CD4-tailed variants from native protein.

– We agree that co-expression experiments would be an obvious choice. We chose not to pursue this as we frequently find internalization experiments to lack consistency when performing co-transfection in the context of over-expression in 293-T-CD20 cells. Further, internalization experiments in the context of viral infection were performed, but suffered from low consistency as we are currently lacking mAbs against HCMV vFcRs, which would be a necessary tool for this approach. Further, as indicated by the new Figure 6—figure supplement 1, following a single target in internalization with a humanized mAb is limited by the low surface expression of gH in the context of infection. Overall, we find internalization experiments to be highly sensitive and only conclusive in a low-complexity setting with high antigen expression.

Figure 4 makes further use of CD4 modified gp34 and gp68 to maintain cell surface expression, and the conclusion, "gp68 but not gp34 to significantly reduce binding of FcγRIIIA/B to cell surface immune complexes compared to CD99" is warranted. Again, it seems combination of native proteins with CD4 modified proteins would add important information to the interpretations offered here.

We agree that co-expression would also mimic a more native combination of vFcR expression. Therefore, we chose to address this directly in the context of viral infection (Figure 5).

Figure 5 returns to virus-infected cells to evaluate "a strong relative reduction in FcγRIIIA/B binding to opsonized HCMV-infected cells in the presence of gp68, but not gp34", which is substantiated by the data shown.Figure 6 addresses the issue of why authors "did not observe efficient antagonization of FcγR activation by gp34 or gp68 individually in the context of HCMV deletion mutant infection". Using an assay set up in HeLa cells, "gp68 or gp34 individually confirmed to antagonize FcγRIII activation", "the more membrane resident gp68-CD4 showed a markedly stronger antagonistic effect compared to gp68wt", and "internalization to be a major condition of gp34 driven antagonization of FcγR activation", which are substantiated. The authors then observe that "the presence of an excess of non-immune IgG interferes with both gp68 and gp34 antagonization" that is "attributed to displacement of immune IgG from the vFcγRs resulting in restoration of FcγRIII activation". All of this data appears useful and appropriate.Figure 7 goes on to show that the "ectodomain of gp68 is sufficient to enhance the antagonistic effect of gp34", which gets at some of the concerns raised with earlier data where combinations would have probably brought this to light.The overall conclusions are warranted, so long as it is clear to the reader that the virus being used lacks all other potentially relevant vFCgRs (when the double mutant is introduced, it would be appropriate to state that it lacks all known vFcgRs). In addition, it would be appropriate to divulge whether elimination of UL12 is necessary for these observations even if studies with conditional RL13 expression are not readily approachable and remains a mystery for future consideration. Limitations of the parental virus backbone should be an important topic for the Discussion. It is unlikely in this reviewer's eye that a set of mutants in fully WT (conditional) virus would behave the same as these particular set are shown to behave, thus discussion relating to clinical issues should be toned down.

We have mentioned this important limitation of our study in the Discussion.

Reviewer #2:The manuscript investigates the mechanism cytomegalovirus proteins gp68 and gp34 as an immune evasion strategy to limit Fc receptor-mediated antibody-dependent cell-mediated cytotoxicity. The study is based on previous findings characterizing gp34 and gp68 as viral evasins during a virus infection. The manuscript describes that gp34 and gp68 function in a synergistic manner by enhances immune complex internalization and as an inhibitor of FcγR binding to immune complexes, respectively. The manuscript systematically examines the function of gP34 and gp68 through virus infection and protein expression in transfected cells. Yet, the manuscript conclusion is based on only two-fold differences with the key conditions and virus studies do not demonstrate synergistic findings (Figure 2).1) Figure 2 is a key experiment that demonstrates that the lack of gp34 and gp68 have an impact on FcyR activation. The data does not show cooperativity to down limit FcyR activation. The study should expand the kinetics of study and different concentrations of Cytotect as well as specific anti-CMV antibodies.

– We thank the reviewer for pointing out a lack of clarity in the presentation of our data. However, we gently disagree regarding the content. Figure 2A and B show a clear over-additive inhibition when gp34 and gp68 are expressed together (blue curve) compared to the isolated expression (red and orange, respectively). The latter curves are close to the level of the ΔΔΔ control. We now included a statistical evaluation of the data comparing “area under curve” in addition to the depicting titration curve.

– We have analysed 3 log10 steps of Cytotect and tested 72 hrs p.i. when gp34 and gp68 are expressed under steady state level conditions (Atalay et al., 2002).

– We performed all experiments with titrated amounts of IVIg and now provide compiled data as “area under curve” graphs for panel A. Titration of IVIg regarding panel B was performed (1 µg and 0.1µg IVIg). We chose to omit this data from the graphs as there was no additional information and NK cell activation remained close to background level.

– The reviewer suggestion to use monoclonal antibodies is very appropriate and was considered by us in the past. Specifically, we performed experiments using MSL-109 as a gH specific humanized monoclonal antibody which is commercially available. Unfortunately, low levels of gH on the surface of HCMV infected cells lead to very low reporter responses despite the high sensitivity of our reporter cell system (see new Figure 6—figure supplement 1). Therefore, we chose to use IVIg in the first place. Fortunately, using IVIg revealed to us the role of non-immune IgG in this setting, which was addressed in Figure 6 in a setting of reduced complexity.

– We conducted a new experiment using a mAb against Her2 (Herceptin) instead of MSL109 in the context of permissive HCMV infection. This was made possible by using BJ foreskin fibroblasts stably expressing Her2. Commercially available humanized anti-HCMV gH (MSL-109) could not be used as we find marginal amounts of gH expressed on the surface of an infected cell.

2) The study is based on previous publications and further define the mechanism of actions. However, the findings typically only demonstrate at most a 2-fold difference upon the expression of gp34 and gp68. This is mostly due to the analysis of using flow cytometry. The authors should include a more sensitive assay to define the function of gp34 and gp68 on FcyR activation.

– We agree that some experiments delineating molecular mechanisms using flow cytometry show weaker effects or when looking at single values along a titration curve. However, the functional read-out systems (e.g. the reporter cell assay as well as primary NK cell degranulation assay) demonstrates robust and significant differences (see Figure 2A and B).